# IMAGE INPAINTING VIA TRACTABLE STEERING OF DIFFUSION MODELS

**Anji Liu**[1]**, Mathias Niepert**[2]**, Guy Van den Broeck**[1]
[1]Department of Computer Science, University of California, Los Angeles
[2]Department of Computer Science, University of Stuttgart
`liuanji@cs.ucla.edu`, `mathias.niepert@simtech.uni-stuttgart.de`,
`guyvdb@cs.ucla.edu`,

## ABSTRACT

Diffusion models are the current state of the art for generating photorealistic images. Controlling the sampling process for constrained image generation tasks such as inpainting, however, remains challenging since exact conditioning on such constraints is intractable. While existing methods use various techniques to approximate the constrained posterior, this paper proposes to exploit the ability of Tractable Probabilistic Models (TPMs) to exactly and efficiently compute the constrained posterior, and to leverage this signal to steer the denoising process of diffusion models. Specifically, this paper adopts a class of expressive TPMs termed Probabilistic Circuits (PCs). Building upon prior advances, we further scale up PCs and make them capable of guiding the image generation process of diffusion models. Empirical results suggest that our approach can consistently improve the overall quality and semantic coherence of inpainted images across three natural image datasets (i.e., CelebA-HQ, ImageNet, and LSUN) with only $\sim 10\%$ additional computational overhead brought by the TPM. Further, with the help of an image encoder and decoder, our method can readily accept semantic constraints on specific regions of the image, which opens up the potential for more controlled image generation tasks. In addition to proposing a new framework for constrained image generation, this paper highlights the benefit of more tractable models and motivates the development of expressive TPMs.

## 1 INTRODUCTION

Thanks to their expressiveness, diffusion models have achieved state-of-the-art results in generating photorealistic and high-resolution images (Ramesh et al., 2022; Nichol & Dhariwal, 2021; Rombach et al., 2022). However, steering unconditioned diffusion models toward constrained generation tasks such as image inpainting remains challenging, as diffusion models do not by design support efficient computation of the posterior sample distribution under many types of constraints (Chung et al., 2022). This results in samples that fail to properly align with the constraints. For example, in image inpainting, the model may generate samples that are semantically incoherent with the given pixels.

Prior works approach this problem mainly by approximating the (constrained) posterior sample distribution. However, due to the intractable nature of diffusion models, such approaches introduce high bias (Lugmayr et al., 2022; Zhang et al., 2023a; Chung et al., 2022) to the sampling process, which diminishes the benefit of using highly-expressive diffusion models.

Having observed that the lack of tractability hinders us from fully exploiting diffusion models in constrained generation tasks, we study the converse problem: *what is the benefit of models that by design support efficient constrained generation?* This paper presents positive evidence by showing that Probabilistic Circuits (PCs) (Choi et al., 2020), a class of expressive Tractable Probabilistic Models that support efficient computation of arbitrary marginal probabilities, can efficiently steer the denoising process of diffusion models towards high-quality inpainted images. We will define a class of constraints that includes inpainting constraints for which we can provide the following guarantee. For any constraint $c$ in this class, given a sample $\boldsymbol{x}_t$ at noise level $t$, we show that a PC trained on noise-free samples (i.e., $p(\mathbf{X}_0)$) can be used to efficiently compute $p(\boldsymbol{x}_0|\boldsymbol{x}_t, c)$, which is a key step in the sampling process of diffusion models. This PC-computed distribution can then

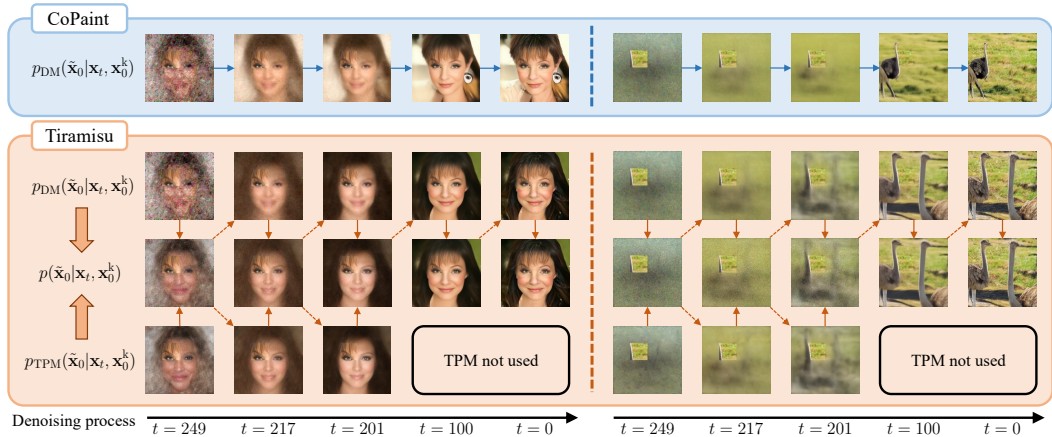

Figure 1: Illustration of the steering effect of the TPM on the diffusion model. The same random seed is used by the baseline (CoPaint; Zhang et al. (2023a)) and our approach. At every time step, given the image at the previous noise level, Tiramisu reconstructs $\tilde{\boldsymbol{x}}_0$ with both the diffusion model and the TPM, and combines the two distributions by taking their geometric mean (solid arrows). The images then go through the noising process to generate the input for the previous time step (dashed arrows).

be used to effectively guide the denoising process, leading to photorealistic images that adhere to the constraints. Figure 1 illustrates the steering effect of PCs in the proposed algorithm **Tiramisu** (**Tra**ctable **Im**age **I**npainting via **S**teering Diff**u**sion Models). Specifically, we plot the reconstructed image by the diffusion model (the first row of Tiramisu) and the PC (the third row) at five time steps during the denoising process. Compared to the baselines, Tiramisu generates more semantically coherent images with the PC-provided guidance. In summary, this paper has three main contributions:

*A new scheme for controlled image generation.* This is the first paper that demonstrates the possibility of using TPMs for controlling/constraining the generation process of natural and high-resolution images. This not only opens up new avenues for controlled image generation, but also highlights the potential impact of non-standard learning architectures (e.g., PCs) on modern image generation tasks.

*Competitive sample quality and runtime.* Empirical evaluations on three challenging high-resolution natural image datasets (i.e., CelebA-HQ, ImageNet, and LSUN) show that the proposed method Tiramisu consistently improves the overall quality of inpainted images while introducing only $\sim 10\%$ additional computational overhead, which is the joint effort of (i) further scaling up PC models based on prior art, and (ii) an improved custom GPU implementation for PC training and inference.

*Potential for more complex controlled generation tasks.* In its more general form, independent soft-evidence constraints include tasks beyond image inpainting. As an illustrative example, we demonstrate that Tiramisu is capable of fusing the semantics of image patches from a set of reference images and generating images conditioned on such semantic constraints. This highlights the potential of Tiramisu on more challenging controlled image generation tasks.

## 2 PRELIMINARIES

**Denoising Diffusion Probabilistic Models**   A diffusion model (Ho et al., 2020; Sohl-Dickstein et al., 2015) defined on variables $\mathbf{X}_0$ is a latent variable model of the form $p_\theta(\boldsymbol{x}_0) := \int p_\theta(\boldsymbol{x}_{0:T}) d\boldsymbol{x}_{1:T}$, where $\boldsymbol{x}_{1:T}$ are the latent variables and the joint distribution $p_\theta(\boldsymbol{x}_{0:T})$ is defined as a Markov chain termed the reverse/denoise process:

$$p_\theta(\boldsymbol{x}_{0:T}) := p(\boldsymbol{x}_T) \cdot \prod_{t=1}^{T} p_\theta(\boldsymbol{x}_{t-1}|\boldsymbol{x}_t). \tag{1}$$

For continuous variables $\boldsymbol{x}_{0:T}$, the initial and transition probabilities of the Markov chain typically use Gaussian distributions: $p(\boldsymbol{x}_T) := \mathcal{N}(\boldsymbol{x}_T; \mathbf{0}, \mathbf{I})$, $p_\theta(\boldsymbol{x}_{t-1}|\boldsymbol{x}_t) := \mathcal{N}(\boldsymbol{x}_{t-1}; \boldsymbol{\mu}_\theta(\boldsymbol{x}_t, t), \boldsymbol{\Sigma}_\theta(\boldsymbol{x}_t, t))$, where $\boldsymbol{\mu}_\theta$ and $\boldsymbol{\Sigma}_\theta$ are the mean and covariance parameters, respectively. The key property that distinguishes diffusion models from other latent variable models such as hierarchical Variational Autoencoders (Vahdat & Kautz, 2020) is the fact that they have a prespecified approximate posterior $q(\boldsymbol{x}_{1:T}|\boldsymbol{x}_0) := \prod_{t=1}^{T} q(\boldsymbol{x}_t|\boldsymbol{x}_{t-1})$. This is called the forward or diffusion process. For continuous variables, the transition probabilities are also defined as Gaussians: $q(\boldsymbol{x}_t|\boldsymbol{x}_{t-1}) :=$

$\mathcal{N}(\boldsymbol{x}_t; \sqrt{1-\beta_t}\boldsymbol{x}_{t-1}, \beta_t\mathbf{I})$, where $\{\beta_t\}_{t=1}^T$ is a noise schedule. Training is done by maximizing the ELBO of $p_\theta(\boldsymbol{x}_0)$ with the variational posterior $q(\boldsymbol{x}_{1:T}|\boldsymbol{x}_0)$. See Kingma et al. (2021) for more details.

While it is possible to directly model $p_\theta(\boldsymbol{x}_{t-1}|\boldsymbol{x}_t)$ with a neural network, prior works discovered that the following parameterization leads to better empirical performance (Ho et al., 2020):

$$p_\theta(\boldsymbol{x}_{t-1}|\boldsymbol{x}_t) := \sum\nolimits_{\tilde{\boldsymbol{x}}_0} q(\boldsymbol{x}_{t-1}|\tilde{\boldsymbol{x}}_0, \boldsymbol{x}_t) \cdot p_\theta(\tilde{\boldsymbol{x}}_0|\boldsymbol{x}_t), \tag{2}$$

where $p_\theta(\tilde{\boldsymbol{x}}_0|\boldsymbol{x}_t) := \mathcal{N}(\tilde{\boldsymbol{x}}_0; \tilde{\boldsymbol{\mu}}_\theta(\boldsymbol{x}_t, t), \tilde{\boldsymbol{\Sigma}}_\theta(\boldsymbol{x}_t, t))$ is parameterized by a neural network and $q(\boldsymbol{x}_{t-1}|\tilde{\boldsymbol{x}}_0, \boldsymbol{x}_t)$ has a simple closed-form expression (Ho et al., 2020). Following the definition of the denoising process, sampling from a diffusion model boils down to first sampling from $p(\boldsymbol{x}_T)$ and then recursively sampling $\boldsymbol{x}_{T-1}, \ldots, \boldsymbol{x}_0$ according to $p_\theta(\boldsymbol{x}_{t-1}|\boldsymbol{x}_t)$.

**Tractable Probabilistic Models**   Tractable Probabilistic Models (TPMs) are a class of generative models that by design support efficient and exact computation of certain queries (Poon & Domingos, 2011; Kisa et al., 2014; Choi et al., 2020; Correia et al., 2023; Sidheekh et al., 2023; Rahman et al., 2014; Kulesza et al., 2012). Depending on their structure, TPMs support various queries ranging from marginal/conditional probabilities to conditioning on logical constraints (Vergari et al., 2021; Bekker et al., 2015). Thanks to their tractability, TPMs enable a wide range of downstream applications such as constrained language generation (Zhang et al., 2023b), knowledge graph link prediction (Loconte et al., 2023), and data compression (Liu et al., 2022a).

## 3    GUIDING DIFFUSION MODELS WITH TRACTABLE PROBABILISTIC MODELS

Given a diffusion model trained for unconditional generation, our goal is to steer the model to generate samples given different conditions/constraints without the need for task-specific fine-tuning. In the following, we focus on the image inpainting task to demonstrate that TPMs can guide diffusion models toward more coherent samples that satisfy the constraints.

The goal of image inpainting is to predict the missing pixels given the known pixels. Define $\mathbf{X}_0^{\mathrm{k}}$ (resp. $\mathbf{X}_0^{\mathrm{u}}$) as the provided (resp. missing) pixels. We aim to enforce the inpainting constraint $\mathbf{X}_0^{\mathrm{k}} = \boldsymbol{x}_0^{\mathrm{k}}$ on every denoising step $p_\theta(\boldsymbol{x}_{t-1}|\boldsymbol{x}_t)$ ($\forall t \in 1, \ldots, T$). Plugging in Equation (2), the conditional probabilities are written as:

$$\forall t \in 1, \ldots, T \quad p_\theta(\boldsymbol{x}_{t-1}|\boldsymbol{x}_t, \boldsymbol{x}_0^{\mathrm{k}}) = \sum\nolimits_{\tilde{\boldsymbol{x}}_0} q(\boldsymbol{x}_{t-1}|\tilde{\boldsymbol{x}}_0, \boldsymbol{x}_t) \cdot p_\theta(\tilde{\boldsymbol{x}}_0|\boldsymbol{x}_t, \boldsymbol{x}_0^{\mathrm{k}}),$$

where the first term on the right-hand side is independent of $\boldsymbol{x}_0^{\mathrm{k}}$. To sample coherent inpainted images, we need to draw unbiased samples from $p_\theta(\tilde{\boldsymbol{x}}_0|\boldsymbol{x}_t, \boldsymbol{x}_0^{\mathrm{k}})$. Applying Bayes' rule, we get

$$p_\theta(\tilde{\boldsymbol{x}}_0|\boldsymbol{x}_t, \boldsymbol{x}_0^{\mathrm{k}}) = \frac{1}{Z} \cdot p_\theta(\tilde{\boldsymbol{x}}_0|\boldsymbol{x}_t) \cdot p(\boldsymbol{x}_0^{\mathrm{k}}|\tilde{\boldsymbol{x}}_0) = \frac{1}{Z} \cdot p_\theta(\tilde{\boldsymbol{x}}_0|\boldsymbol{x}_t) \cdot \mathbb{1}[\tilde{\boldsymbol{x}}_0^{\mathrm{k}} = \boldsymbol{x}_0^{\mathrm{k}}], \tag{3}$$

where $\mathbb{1}[\cdot]$ is the indicator function and $Z$ is a normalizing constant. One simple strategy to sample from Equation (3) is by rejection sampling: sample unconditionally from $p_\theta(\tilde{\boldsymbol{x}}_0|\boldsymbol{x}_t)$ (i.e., the learned denoising model) and reject samples with $\tilde{\boldsymbol{x}}_0^{\mathrm{k}} \neq \boldsymbol{x}_0^{\mathrm{k}}$. However, this is impractical since the acceptance rate could be extremely low. Existing algorithms use approximation strategies to compute or sample from Equation (3). For example, Lugmayr et al. (2022) proposes to set $\tilde{\boldsymbol{x}}_0^{\mathrm{k}} := \boldsymbol{x}_0^{\mathrm{k}}$ after sampling $\tilde{\boldsymbol{x}}_0 \sim p_\theta(\cdot|\boldsymbol{x}_t)$, leaving $\tilde{\boldsymbol{x}}_0^{\mathrm{u}}$ untouched; Zhang et al. (2023a) and Chung et al. (2022) relax the hard inpainting constraint to $\min_{\tilde{\boldsymbol{x}}_0^{\mathrm{k}}} \|\tilde{\boldsymbol{x}}_0^{\mathrm{k}} - \boldsymbol{x}_0^{\mathrm{k}}\|_2^2$ and use gradient-based methods to gradually enforce it.

This paper explores the possibility of drawing unbiased samples from $p_\theta(\tilde{\boldsymbol{x}}_0|\boldsymbol{x}_t, \boldsymbol{x}_0^{\mathrm{k}})$ given a TPM-represented distribution $p_\theta(\boldsymbol{x}_0)$. By applying Bayes' rule from the other side, we have

$$p_\theta(\tilde{\boldsymbol{x}}_0|\boldsymbol{x}_t, \boldsymbol{x}_0^{\mathrm{k}}) = \frac{1}{Z} \cdot q(\boldsymbol{x}_t|\tilde{\boldsymbol{x}}_0) \cdot p_\theta(\tilde{\boldsymbol{x}}_0|\boldsymbol{x}_0^{\mathrm{k}}) = \frac{1}{Z} \cdot \prod_i q(x_t^i|\tilde{x}_0^i) \cdot p_\theta(\tilde{\boldsymbol{x}}_0^{\mathrm{u}}|\boldsymbol{x}_0^{\mathrm{k}}) \cdot \mathbb{1}[\tilde{\boldsymbol{x}}_0^{\mathrm{k}} = \boldsymbol{x}_0^{\mathrm{k}}], \tag{4}$$

where $Z$ is a normalizing constant, $x_t^i$ is the $i$th variable in $\boldsymbol{x}_t$, and the factorization of $q(\boldsymbol{x}_t|\boldsymbol{x}_0)$ follows the definition of the diffusion process in Section 2. Although the right-hand side seems to be impractical to compute due to the normalizing constant, we will demonstrate in the following sections that there exists a class of expressive TPMs that can compute it efficiently and exactly.

From the diffusion model and the TPM, we have obtained two versions of the same distribution $p(\tilde{\boldsymbol{x}}_0|\boldsymbol{x}_t, \boldsymbol{x}_0^{\mathrm{k}})$.[1] Thanks to the expressiveness of neural networks, the distribution approximated by the diffusion model (i.e., Eq. (3); termed $p_{\mathrm{DM}}(\tilde{\boldsymbol{x}}_0|\boldsymbol{x}_t, \boldsymbol{x}_0^{\mathrm{k}})$) encodes high-fidelity images. However, due

---

[1]Note that diffusion models can only approximate this distribution.

to the inability to compute the exact conditional probability, $p_{\text{DM}}(\tilde{\boldsymbol{x}}_0|\boldsymbol{x}_t, \boldsymbol{x}_0^{\text{k}})$ could lead to images incoherent with the constraint (Zhang et al., 2023a). In contrast, the TPM-generated distribution (i.e., Eq. (4); termed $p_{\text{TPM}}(\tilde{\boldsymbol{x}}_0|\boldsymbol{x}_t, \boldsymbol{x}_0^{\text{k}})$) represents images that potentially better align with the given pixels. Therefore, $p_{\text{DM}}(\tilde{\boldsymbol{x}}_0|\boldsymbol{x}_t, \boldsymbol{x}_0^{\text{k}})$ and $p_{\text{TPM}}(\tilde{\boldsymbol{x}}_0|\boldsymbol{x}_t, \boldsymbol{x}_0^{\text{k}})$ can be viewed as distributions trained for the same task yet with different biases. Following prior arts (Grover & Ermon, 2018; Zhang et al., 2023b), we combine both distributions by taking the weighted average of the logits of every variable in $\tilde{\boldsymbol{x}}_0$, hoping to get images that are both semantically coherent and have high fidelity:

$$p(\tilde{\boldsymbol{x}}_0|\boldsymbol{x}_t, \boldsymbol{x}_0^{\text{k}}) \propto p_{\text{DM}}(\tilde{\boldsymbol{x}}_0|\boldsymbol{x}_t, \boldsymbol{x}_0^{\text{k}})^{\alpha} \cdot p_{\text{TPM}}(\tilde{\boldsymbol{x}}_0|\boldsymbol{x}_t, \boldsymbol{x}_0^{\text{k}})^{1-\alpha}, \tag{5}$$

where $\alpha \in (0, 1)$ is a mixing hyperparameter. In summary, as a key step of image inpainting with diffusion models, we compute $p(\tilde{\boldsymbol{x}}_0|\boldsymbol{x}_t, \boldsymbol{x}_0^{\text{k}})$ from both the diffusion model and a TPM, and use their weighted geometric mean in the denoising process. We note that the use of TPMs is independent of the design choices related to the diffusion model, and thus can be built upon any prior approach.

## 4 PRACTICAL IMPLEMENTATION WITH PROBABILISTIC CIRCUITS

The previous section introduces how TPMs could help guide the denoising process of diffusion models toward high-quality inpainted images. While promising, a key question is *whether* $p_{\text{TPM}}(\tilde{\boldsymbol{x}}_0|\boldsymbol{x}_t, \boldsymbol{x}_0^{\text{k}})$ *(Eq. 4) can be computed efficiently and exactly?* We answer the question in its affirmative by showing that a class of TPMs termed Probabilistic Circuits (PCs) (Choi et al., 2020) can answer the query while being expressive enough to model natural images. In the following, we first provide background on PCs (Sec. 4.1). We then describe how they are used to compute $p_{\text{TPM}}(\tilde{\boldsymbol{x}}_0|\boldsymbol{x}_t, \boldsymbol{x}_0^{\text{k}})$ (Sec. 4.2).

### 4.1 BACKGROUND ON PROBABILISTIC CIRCUITS

Probabilistic Circuits (PCs) (Choi et al., 2020) are an umbrella term for a wide variety of TPMs, including classic ones such as Hidden Markov Models (Rabiner & Juang, 1986) and Chow-Liu Trees (Chow & Liu, 1968) as well as more recent ones including Sum-Product Networks (Poon & Domingos, 2011), Arithmetic Circuits (Shen et al., 2016), and Cutset Networks (Rahman et al., 2014). We define the syntax and semantics of PCs as follows.

**Definition 1** (Probabilistic Circuits). A PC $p(\mathbf{X})$ represents a distribution over $\mathbf{X}$ via a parameterized Directed Acyclic Graph (DAG) with a single root node $n_{\text{r}}$. There are three types of nodes in the DAG: *input*, *product*, and *sum* nodes. Input nodes define primitive distributions over some variable $X \in \mathbf{X}$, while sum and product nodes merge the distributions defined by their children, denoted $\text{in}(n)$, to build more complex distributions. Specifically, the distribution encoded by every node is defined recursively as

Figure 2: An example PC over boolean variables $X_1, \ldots, X_4$. Sum parameters are labeled on the corresponding edges. The probability of every node given input $x_1\bar{x}_2\bar{x}_3x_4$ is labeled blue on top of the corresponding node.

$$p_n(\boldsymbol{x}) := \begin{cases} f_n(\boldsymbol{x}) & n \text{ is an input node,} \\ \prod_{c \in \text{in}(n)} p_c(\boldsymbol{x}) & n \text{ is a product node,} \\ \sum_{c \in \text{in}(n)} \theta_{n,c} \cdot p_c(\boldsymbol{x}) & n \text{ is a sum node,} \end{cases} \tag{6}$$

where $f_n(\boldsymbol{x})$ is an univariate input distribution (e.g., Gaussian, Categorical), and $\theta_{n,c}$ denotes the parameter corresponds to edge $(n, c)$. Intuitively, sum nodes and product nodes encode mixture and factorized distributions of their children, respectively. To ensure that a PC models a valid distribution, we assume the child parameters of every sum node $n$ (i.e., $\{\theta_{n,c}\}_{c \in \text{in}(n)}$) sum up to 1. The size of a PC $p$, denoted $|p|$, is the number of edges in its DAG.

Figure 2 shows an example PC over boolean variables $X_1, \ldots, X_4$, where $\odot$, $\otimes$, and $\oplus$ represent input, product, and sum nodes, respectively. The key to guaranteeing the tractability of PCs is to add proper structural constraints to their DAG structure. Specifically, this paper considers smoothness and decomposability, which are required by the inference algorithm that will be introduced in Section 4.2.

**Definition 2** (Smoothness and Decomposability). Define the scope $\phi(n)$ of node $n$ as the collection of variables defined by all its descendent input nodes. A PC is smooth if for every sum node $n$, its children have the same scope: $\forall c_1, c_2 \in \text{in}(n), \phi(c_1) = \phi(c_2)$; it is decomposable if for every product node $n$, its children have disjoint scopes: $\forall c_1, c_2 \in \text{in}(n) \ (c_1 \neq c_2), \phi(c_1) \cap \phi(c_2) = \varnothing$.

Answering queries with PCs amounts to computing certain functions recursively in postorder (i.e., feedforward) or preorder (i.e., backward) on its DAG. For example, computing the likelihood $p(\boldsymbol{x})$ boils down to a forward pass on the PC: we first assign a probability to every input node $n$ by evaluating its density/mass function $f_n(\boldsymbol{x})$, and then do a feedforward pass (children before parents) of all sum and product nodes, computing their output probabilities following Equation (6). Finally, the output value of the root node is the target likelihood. In Figure 2, the output probability of every node for the query $p(x_1\bar{x}_2\bar{x}_3x_4)$ is labeled blue on top of it.

As hinted by its definition, the set of learnable parameters in PCs includes (i) parameters of the sum edges and (ii) parameters of the input nodes/distributions. All parameters can be jointly learned using an Expectation-Maximization-based algorithm that aims to maximize the average log-likelihood of all samples in a dataset $\mathcal{D}$: $\sum_{\boldsymbol{x}\in\mathcal{D}}\log p_r(\boldsymbol{x})$. Details of the EM algorithm is provided in Appendix C.1.

## 4.2 Computing Constrained Posterior Distribution

Recall from Section 3 and Equation (5) that at every denoising step, we need to compute $p_{\text{TPM}}(\tilde{\boldsymbol{x}}_0|\boldsymbol{x}_t,\boldsymbol{x}_0^{\text{k}})$ with the PC. This section proposes an algorithm that computes $p_{\text{TPM}}(\tilde{\boldsymbol{x}}_0|\boldsymbol{x}_t,\boldsymbol{x}_0^{\text{k}})$ given a PC $p(\boldsymbol{x}_0)$ in linear time w.r.t. its size. Specifically, we first demonstrate how Equation (4) can be converted to a general form of queries we define as *independent soft-evidence constraints*. We then establish an efficient inference algorithm for this query class.

After closer inspection of Equation (4), we observe that both $q(\boldsymbol{x}_t|\tilde{\boldsymbol{x}}_0)$ and $\mathbb{1}[\tilde{\boldsymbol{x}}_0^{\text{k}}=\boldsymbol{x}_0^{\text{k}}]$ can be considered as constraints factorized over every variable. Specifically, with $w_i(\tilde{x}_0^i):=q(x_t^i|\tilde{x}_0^i)$ if $\tilde{X}_0^i\in\tilde{\mathbf{X}}_0^{\text{u}}$ and $w_i(\tilde{x}_0^i):=\mathbb{1}[\tilde{x}_0^{\text{k}}=x_0^{\text{k}}]$ otherwise, $p_{\text{TPM}}(\tilde{\boldsymbol{x}}_0|\boldsymbol{x}_t,\boldsymbol{x}_0^{\text{k}})$ can be equivalently expressed as

$$p_{\text{TPM}}(\tilde{\boldsymbol{x}}_0|\boldsymbol{x}_t,\boldsymbol{x}_0^{\text{k}}) = \frac{1}{Z}\prod_i w_i(\tilde{x}_0^i)\cdot p(\tilde{\boldsymbol{x}}_0), \text{ where } Z := \sum_{\boldsymbol{x}_0}\prod_i w_i(\tilde{x}_0^i)\cdot p(\tilde{\boldsymbol{x}}_0). \tag{7}$$

We call $w_i$ the *soft-evidence constraint* of variable $X_0^i$ as it defines a prior belief of its value. In the extreme case of conditioning on hard evidence, $w_i$ becomes an indicator that puts all weight on the conditioned value. Recall from Section 2 that diffusion models parameterize $p_\theta(\tilde{\boldsymbol{x}}_0|\boldsymbol{x}_t)$ as a fully-factorized distribution. In order to compute the weighted geometric mean of $p_{\text{DM}}(\tilde{\boldsymbol{x}}_0|\boldsymbol{x}_t,\boldsymbol{x}_0^{\text{k}})$ and $p_{\text{TPM}}(\tilde{\boldsymbol{x}}_0|\boldsymbol{x}_t,\boldsymbol{x}_0^{\text{k}})$ (cf. Eq. (5)), we need to also compute the univariate distributions $p_{\text{TPM}}(\tilde{x}_0^i|\boldsymbol{x}_t,\boldsymbol{x}_0^{\text{k}})$ for every $\tilde{X}_0^i\in\tilde{\mathbf{X}}_0$. While this seems to suggest the need to query the PC at least $|\tilde{\mathbf{X}}_0|$ times, we propose an algorithm that only needs a forward and a backward pass to compute *all* target probabilities.

**The forward pass** Similar to the likelihood query algorithm introduced in Section 4.1, we traverse all nodes in postorder and store the output of every node $n$ in $\texttt{fw}_n$. For sum and product nodes, the output is computed following Equation (6); the output of every input node $n$ that encodes a distribution of $X_0^i$ is defined as $\texttt{fw}_n := \sum_{x_0^i} f_n(x_0^i)\cdot w_i(x_0^i)$, where $f_n$ is defined in Equation (6).

**The backward pass** The backward pass consists of two steps: (i) traversing all nodes in preorder (parents before children) to compute the backward value $\texttt{bk}_n$; (ii) computing the target probabilities using the backward value of all input nodes. For ease of presentation, we assume the PC alternates between sum and product layers, and all parents of any input node are product nodes.[2] First, we compute the backward values by setting $\texttt{bk}_{n_\text{r}}$ of the root node to 1, then recursively compute the backward value of other nodes as follows:

$$\texttt{bk}_n := \begin{cases} \sum_{m\in\texttt{pa}(n)} \left(\theta_{m,n}\cdot\texttt{fw}_n/\texttt{fw}_m\right)\cdot\texttt{bk}_m & n \text{ is a product node,} \\ \sum_{m\in\texttt{pa}(n)}\texttt{bk}_m & n \text{ is a input or sum node,} \end{cases}$$

where $\texttt{pa}(n)$ is the set of parents of node $n$. Next, for every $i$, we gather all input nodes defined on $X_0^i$, denoted $S_i$, and compute $p_{\text{TPM}}(\tilde{x}_0^i|\boldsymbol{x}_t,\boldsymbol{x}_0^{\text{k}}) := \frac{1}{Z}\sum_{n\in S_i}\texttt{bk}_n\cdot f_n(x_0^i)\cdot w_i(x_0^i)$. We justify the correctness of this algorithm in the following theorem, whose proof is provided in Appendix A.

**Theorem 1.** *For any smooth and decomposable PC $p(\mathbf{X})$ and univariate weight functions $\{w_i(X_i)\}_i$, define $p'(\boldsymbol{x}) = \frac{1}{Z}\prod_i w_i(x_i)\cdot p(\boldsymbol{x})$, where the normalizing constant $Z := \sum_{\boldsymbol{x}}\prod_i w_i(x_i)\cdot p(\boldsymbol{x})$. Assume all variables in $\mathbf{X}$ are categorical variables with $C$ categories, the above-described algorithm computes $p'(x_i)$ for every variable $X_i$ and its every assignment $x_i$ in time $\mathcal{O}(|p|+|\mathbf{X}|\cdot C)=\mathcal{O}(|p|)$.*

---

[2] Every PC that does not satisfy such constraints can be transformed into one in linear time since (i) consecutive sum or product nodes can be merged without changing the PC's semantic, and (ii) we can add a dummy product with one child between any pair of sum and input nodes.

## 5 Towards High-Resolution Image Inpainting

Another key factor determining the effectiveness of the PC-guided diffusion model is the expressiveness of the PC $p(\mathbf{X}_0)$, i.e., how well it can model the target image distribution. Recent advances have significantly pushed forward the expressiveness of PCs (Liu et al., 2022b; 2023), leading to competitive likelihoods on datasets such as CIFAR (Krizhevsky et al., 2009) and down-sampled ImageNet (Deng et al., 2009), which allows us to directly apply the guided inpainting algorithm to them. However, there is still a gap towards directly modeling high-resolution (e.g., $256 \times 256$) image data. While it is possible that this could be achieved in the near future given the rapid development of PCs, this paper explores an alternative approach where we use a (variational) auto-encoder to transform high-resolution images to a lower-dimensional latent space. Although in this way we lose the "full tractability" over every pixel, as we shall proceed to demonstrate, a decent approximation can still be achieved. The key intuition is that the latent space concisely captures the semantic information of the image, and thus can effectively guide diffusion models toward generating semantically coherent images; fine-grained details such as color consistency of the neighboring pixels can be properly handled by the neural-network-based diffusion model. This is empirically justified in Section 6.1.

Define the latent space of the image $\mathbf{X}_0$ as $\mathbf{Z}_0$. We adopt Vector Quantized Generative Adversarial Networks (VQ-GANs) (Esser et al., 2021), which are equipped with an encoder $q(\boldsymbol{z}_0|\boldsymbol{x}_0)$ and a decoder $p(\boldsymbol{x}_0|\boldsymbol{z}_0)$, to transform the images between the pixel space and the latent space.[3] We approximate $p_{\text{TPM}}(\tilde{\boldsymbol{x}}_0|\boldsymbol{x}_t, \boldsymbol{x}_0^k)$ (Eq. 7) by first estimating $p_{\text{TPM}}(\tilde{\boldsymbol{z}}_0|\boldsymbol{x}_t, \boldsymbol{x}_0^k)$ with a PC $p(\mathbf{Z}_0)$ trained on the latent space and the VQ-GAN encoder; this latent-space conditional distribution is then converted back to the pixel space. Specifically, the latent-space conditional probability is approximated via

$$p_{\text{TPM}}(\tilde{\boldsymbol{z}}_0|\boldsymbol{x}_t, \boldsymbol{x}_0^k) \approx \frac{1}{Z} \prod_i w_i^{\text{z}}(\tilde{z}_0^i) \cdot p(\tilde{\boldsymbol{z}}_0), \text{ where } w_i^{\text{z}}(\tilde{z}_0^i) := \frac{1}{Z_i} \sum_{\tilde{\boldsymbol{x}}_0} \prod_j w_j(\tilde{x}_0^j) \cdot q(\tilde{z}_0^i|\tilde{\boldsymbol{x}}_0), \quad (8)$$

where $Z$ and $\{Z_i\}_i$ are normalizing constants and $q(\tilde{z}_0^i|\tilde{\boldsymbol{x}}_0)$ is the VQ-GAN encoder. It is safe to assume the independence between the soft evidence for different latent variables (i.e., $w_i^{\text{z}}$) since every latent variable produced by VQ-GAN corresponds to a different image patch, which corresponds to a different set of pixel-space soft constraints (i.e., $w_i$). In practice, we approximate $w_i^{\text{z}}(\tilde{z}_0^i)$ by performing Monte Carlo sampling over $\tilde{\boldsymbol{x}}_0$ (i.e., sample $\tilde{\boldsymbol{x}}_0$ following $\prod_j w_j(\tilde{x}_0^j)$, and then feed them through the VQ-GAN encoder). Finally, $p_{\text{TPM}}(\tilde{\boldsymbol{x}}_0|\boldsymbol{x}_t, \boldsymbol{x}_0^k)$ is approximated by Monte Carlo estimation of $p_{\text{TPM}}(\tilde{\boldsymbol{x}}_0|\boldsymbol{x}_t, \boldsymbol{x}_0^k) := \mathbb{E}_{\tilde{\boldsymbol{z}}_0 \sim p_{\text{TPM}}(\cdot|\boldsymbol{x}_t, \boldsymbol{x}_0^k)}[p(\tilde{\boldsymbol{x}}_0|\tilde{\boldsymbol{z}}_0)]$, where $p(\tilde{\boldsymbol{x}}_0|\tilde{\boldsymbol{z}}_0)$ is the VQ-GAN decoder. We observe that as few as 4-8 samples lead to significant performance gain across various datasets and mask types. See Appendix B for details of the design choices.

Another main contribution of this paper is to further scale up PCs based on Liu et al. (2022b; 2023) to achieve likelihoods competitive with GPTs (Brown et al., 2020) on the latent image space generated by VQ-GAN. Specifically, for $256 \times 256$ images, the latent space typically consists of $16 \times 16 = 256$ categorical variables each with 2048-16384 categories. While the number of variables is similar to datasets considered by prior PC learning approaches, the variables are much more semantically complicated (e.g., patch semantic vs. pixel value). We provide the full learning details including the model structure and the training pipeline in Appendix C.2.

In summary, similar to the pixel-space guided inpainting algorithm introduced in Section 3 and 4.2, its latent-space variant also computes $p_{\text{TPM}}(\tilde{\boldsymbol{x}}_0|\boldsymbol{x}_t, \boldsymbol{x}_0^k)$ to guide the diffusion model $p_{\text{DM}}(\tilde{\boldsymbol{x}}_0|\boldsymbol{x}_t, \boldsymbol{x}_0^k)$ with Equation (5), except that it is approximated using a latent-space PC combined with VQ-GAN.

## 6 Experiments

In this section, we take gradual steps to analyze and illustrate our method **Tiramisu** (**Tra**ctable **Im**age **In**painting via **S**teering Diff**u**sion Models). Specifically, we first qualitatively investigate the steering effect of the TPM on the denoising diffusion process (Sec. 6.1). Next, we perform an empirical evaluation on three high-resolution image datasets with six large-hole masks, which significantly challenges its ability to generate semantically consistent images (Sec. 6.2). Finally, inspired by the fact that Tiramisu can handle arbitrary constraints that can be written as independent soft evidence (cf. Sec. 4.2), we test it on a new controlled image generation task termed *image semantic fusion*, where the goal is to fuse parts from different images and generate images with semantic coherence and high fidelity (Sec. 6.3). Code is available at `https://github.com/UCLA-StarAI/Tiramisu`.

---

[3]We adopt VQ-GAN since it has a discrete latent space, which makes PC training easier.

Table 1: Quantative results on three datasets: CelebA-HQ (Liu et al., 2015), ImageNet (Deng et al., 2009), and LSUN-Bedroom (Yu et al., 2015). We report the average LPIPS value (lower is better) (Zhang et al., 2018) across 100 inpainted images for all settings. Bold indicates the best result.

| Tasks | | Algorithms | | | | | | |
|---|---|---|---|---|---|---|---|---|
| Dataset | Mask | Tiramisu (ours) | CoPaint | RePaint | DDNM | DDRM | DPS | Resampling |
| CelebA-HQ | Left | 0.189 | **0.185** | 0.195 | 0.254 | 0.275 | 0.201 | 0.257 |
| | Top | 0.187 | **0.182** | 0.187 | 0.248 | 0.267 | 0.187 | 0.251 |
| | Expand1 | **0.454** | 0.468 | 0.504 | 0.597 | 0.682 | 0.466 | 0.613 |
| | Expand2 | 0.442 | 0.455 | 0.480 | 0.585 | 0.686 | **0.434** | 0.601 |
| | V-strip | **0.487** | 0.502 | 0.517 | 0.625 | 0.724 | 0.535 | 0.647 |
| | H-strip | **0.484** | 0.488 | 0.517 | 0.626 | 0.731 | 0.492 | 0.639 |
| | Wide | **0.069** | 0.072 | 0.075 | 0.112 | 0.132 | 0.078 | 0.128 |
| ImageNet | Left | **0.286** | 0.289 | 0.296 | 0.410 | 0.369 | 0.327 | 0.369 |
| | Top | **0.308** | 0.312 | 0.336 | 0.427 | 0.373 | 0.343 | 0.368 |
| | Expand1 | **0.616** | 0.623 | 0.691 | 0.786 | 0.726 | 0.621 | 0.711 |
| | Expand2 | **0.597** | 0.607 | 0.692 | 0.799 | 0.724 | 0.618 | 0.721 |
| | V-strip | 0.646 | 0.654 | 0.741 | 0.851 | 0.761 | **0.637** | 0.759 |
| | H-strip | 0.657 | 0.660 | 0.744 | 0.851 | 0.753 | **0.647** | 0.774 |
| | Wide | **0.125** | 0.128 | 0.127 | 0.198 | 0.197 | 0.132 | 0.196 |
| LSUN-Bedroom | Left | **0.285** | 0.287 | 0.314 | 0.345 | 0.366 | 0.314 | 0.367 |
| | Top | **0.310** | 0.323 | 0.347 | 0.376 | 0.368 | 0.355 | 0.372 |
| | Expand1 | **0.615** | 0.637 | 0.676 | 0.716 | 0.695 | 0.641 | 0.699 |
| | Expand2 | **0.635** | 0.641 | 0.666 | 0.720 | 0.691 | 0.638 | 0.690 |
| | V-strip | **0.672** | 0.676 | 0.711 | 0.760 | 0.721 | 0.674 | 0.725 |
| | H-strip | 0.679 | 0.686 | 0.722 | 0.766 | 0.726 | **0.674** | 0.724 |
| | Wide | 0.116 | 0.115 | 0.124 | 0.135 | 0.204 | **0.108** | 0.202 |
| Average | | **0.421** | 0.427 | 0.459 | 0.532 | 0.531 | 0.434 | 0.514 |

## 6.1 ANALYSIS OF THE TPM-PROVIDED GUIDANCE

Since we are largely motivated by the ability of TPMs to generate images that better match the semantics of the given pixels, it is natural to examine how the TPM-generated signal guides the diffusion model during the denoising process. Recall from Section 3 that at every denoising step $t$, the reconstruction distributions $p_{\mathrm{DM}}(\tilde{\boldsymbol{x}}_0|\boldsymbol{x}_t, \boldsymbol{x}_0^{\mathrm{k}})$ and $p_{\mathrm{TPM}}(\tilde{\boldsymbol{x}}_0|\boldsymbol{x}_t, \boldsymbol{x}_0^{\mathrm{k}})$ are computed/estimated using the diffusion model and the TPM, respectively. Both distributions are then merged into $p(\tilde{\boldsymbol{x}}_0|\boldsymbol{x}_t, \boldsymbol{x}_0^{\mathrm{k}})$ (Eq. (5)) and are used to generate the image at the previous noise level (i.e., $\boldsymbol{x}_{t-1}$). In all experiments, we adopt CoPaint (Zhang et al., 2023a) to generate $p_{\mathrm{DM}}(\tilde{\boldsymbol{x}}_0|\boldsymbol{x}_t, \boldsymbol{x}_0^{\mathrm{k}})$, which is independent of the design choices related to the TPM. Therefore, qualitatively comparing the denoising process of Tiramisu and CoPaint allows us to examine the steering effect provided by the TPM.

Figure 1 visualizes the denoising process of Tiramisu by plotting the images corresponding to the expected values of the aforementioned distributions (i.e., $p_{\mathrm{DM}}(\tilde{\boldsymbol{x}}_0|\boldsymbol{x}_t, \boldsymbol{x}_0^{\mathrm{k}})$, $p_{\mathrm{TPM}}(\tilde{\boldsymbol{x}}_0|\boldsymbol{x}_t, \boldsymbol{x}_0^{\mathrm{k}})$, and $p(\tilde{\boldsymbol{x}}_0|\boldsymbol{x}_t, \boldsymbol{x}_0^{\mathrm{k}})$). To minimize distraction, we first focus on image pairs of the DM- and TPM-generated image pairs in the same column. Since they are generated from the same input image $\boldsymbol{x}_t$, comparing the image pairs allows us to examine the built-in inductive biases in both distributions. For instance, in the celebrity face image, we observe that the contour of the facial features is sharper for the TPM-generated image. This is more obvious in images at larger time steps since the guidance provided by the TPM is accumulated throughout the denoising process.

Next, we look at the second row (i.e., $p(\tilde{\boldsymbol{x}}_0|\boldsymbol{x}_t, \boldsymbol{x}_0^{\mathrm{k}})$) of Tiramisu. Although blurry, global semantics appear at the early stages of the denoising process. For example, on the right side, we can vaguely see two ostriches visible at time step 217. In contrast, the denoised image at $t = 217$ for CoPaint does not contain much semantic information. Conditioning on these blurred contents, the diffusion model can further fill in fine-grained details. Since the image semantics can be generated in a few denoising steps, we only need to query the TPM at early time steps, which also significantly reduces the computational overhead of Tiramisu. See Section 6.2 for quantitative analysis. As a result, compared to the baseline, Tiramisu can generate inpainted images with higher quality.

## 6.2 COMPARISON WITH THE STATE OF THE ART

In this section, we challenge Tiramisu against state-of-the-art diffusion-based inpainting algorithms on three large-scale high-resolution image datasets: CelebA-HQ (Liu et al., 2015), ImageNet (Deng et al., 2009), and LSUN-Bedroom (Yu et al., 2015). To further challenge the ability of Tiramisu to generate semantically coherent images, we use seven types of masks that reveal only 5-20% of the original image since it is very likely for inpainting algorithms to ignore the given visual cues and generate semantically inconsistent images. Details of the masks can be found in Appendix D.

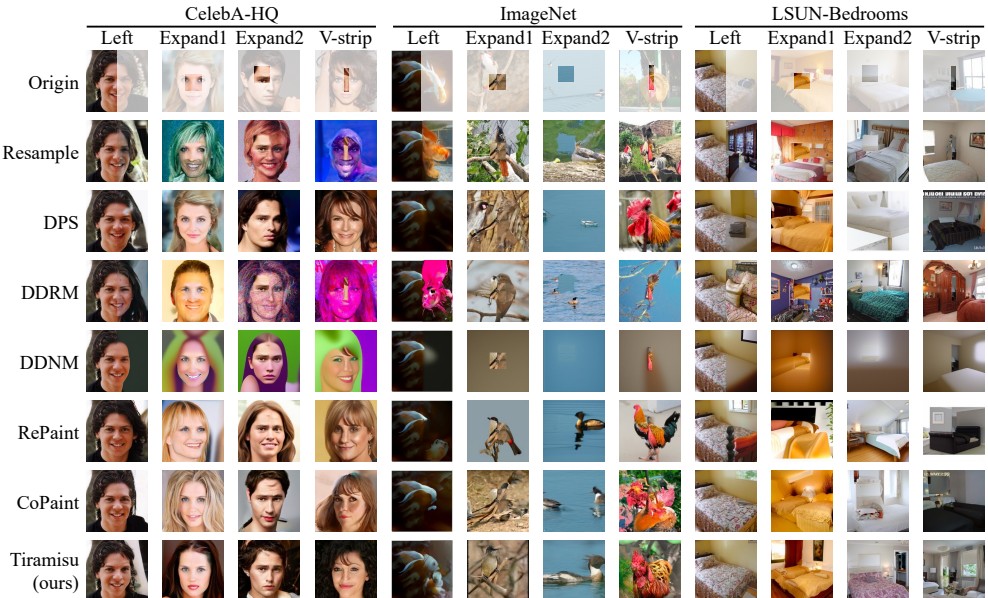

Figure 3: Qualitative results on all three adopted datasets. We compare Tiramisu against six diffusion-based inpainting algorithms. Please refer to Appendix E.2 for more qualitative results.

**Methods** We consider the six following diffusion-based inpainting algorithms: CoPaint (Zhang et al., 2023a), RePaint (Lugmayr et al., 2022), DDNM (Wang et al., 2022), DDRM (Kawar et al., 2022), DPS (Chung et al., 2022), and Resampling (Trippe et al., 2022). Although not exhaustive, this set of methods summarizes recent developments in image inpainting and can be deemed as state-of-the-art. We base our method Tiramisu on CoPaint (i.e., generate $p_{\mathrm{DM}}(\tilde{\boldsymbol{x}}_0|\boldsymbol{x}_t, \boldsymbol{x}_0^{\mathrm{k}})$ with CoPaint). Please see the appendix for details on Tiramisu (Appx. B and C.2) and the baselines (Appx. D).

**Quantitative and qualitative results** Table 1 shows the average LPIPS values (Zhang et al., 2018) on all $3 \times 7 = 21$ dataset-mask configurations. First, Tiramisu outperforms CoPaint in 18 out of 21 settings, which demonstrates that the TPM-provided guidance consistently improves the quality of generated images. Next, compared to all baselines, Tiramisu achieves the best LPIPS value on 14 out of 21 settings, which indicates its superiority over the baselines. This conclusion is further supported by the sample inpainted images shown in Figure 3, which suggests that Tiramisu generates more semantically consistent images. See Appx. E.2 for more samples and Appx. E.1 for user studies.

**Computational efficiency** As illustrated in Section 6.1, we can use PC to steer the denoising steps only in earlier stages. While engaging PCs in more denoising steps could lead to better performance, the runtime is also increased accordingly. To better understand this tradeoff, we use CelebA + the Expand1 mask as an example to analyze this tradeoff. As shown in Figure 4, as we use PCs in more denoising steps, the LPIPS score first decreases and then increases, suggesting that incorporating PCs in a moderate amount of steps gives the best performance (around 20% in this case). One explanation to this phenomenon

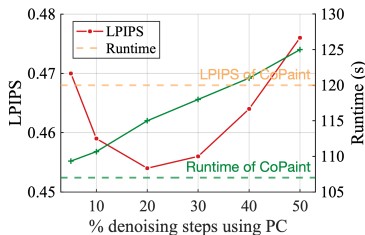

Figure 4: Performance and runtime.

is that in later denoising stages, the diffusion model mainly focuses on refining details. However, PCs are better at controlling the global semantics of images in earlier denoising stages. We then focus on the computation time. When using PCs in 20% of the denoising steps, the additional computational overhead incurred by the TPM is around 10s, which is only 10% of the total computation time.

## 6.3 BEYOND IMAGE INPAINTING

The previous sections demonstrate the effectiveness of using TPMs on image inpainting tasks. A natural follow-up question is *whether this framework can be generalized to other controlled/constrained image generation tasks?* Although we do not have a definite answer, this section demonstrates the potential of extending Tiramisu to more complicated tasks by showing its capability to fuse the semantic information from various input patches/fragments. Specifically, consider the case of

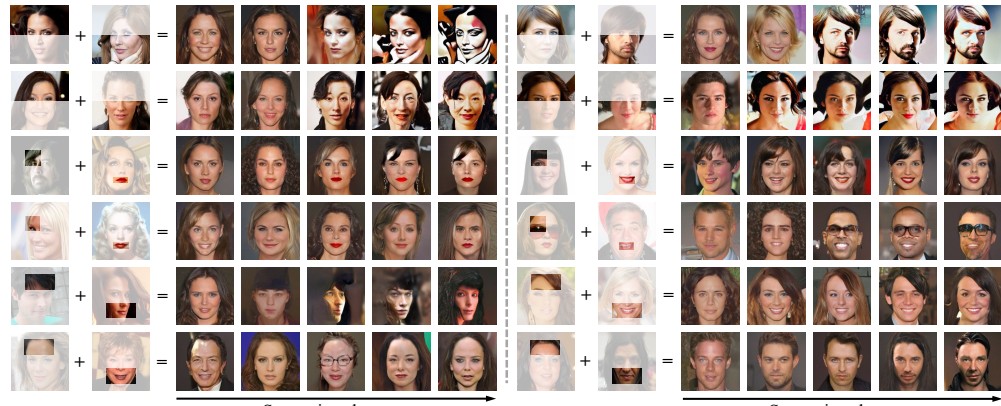

Figure 5: CelebA-HQ qualitative results for the semantic fusion task. In every sample, two reference images together with their masks are provided to Tiramisu. The task is to generate images that (i) semantically align with the unmasked region of both reference images, and (ii) have high fidelity. For every input, we generate five samples with different levels of semantic coherence. The left-most images are the least semantically constrained and barely match the semantic patterns of the reference images. In contrast, the right-most images strictly match the semantics of the reference images.

latent-space soft evidence constraints $\{w_i^z\}_i$ (i.e., Eq. (8)). For various recent autoencoder models such as VQ-GAN, the latent variables of size $H_1 \times W_1$ are encoded from images of size $H \times W$. Intuitively, every latent variable encodes the semantic of an $H/H_1 \times W/W_1$ image patch. Therefore, every $w_i^z$ can be viewed as a constraint on the semantics of the corresponding image patch.

We introduce a controlled image generation task called *semantic fusion*, where we are given several reference images each paired with a mask. The goal is to generate images that (i) semantically align with the unmasked region of every reference image, and (ii) have high quality and fidelity. Semantic fusion can be viewed as a preliminary task for more general controlled image generation since any type of visual word information (e.g., language condition) can be transferred to constraints on $\{w_i^z\}_i$.

Figure 5 shows qualitative results of Tiramisu on semantic fusion tasks. For every set of reference images, we generate five samples with different semantic coherence levels by adjusting the temperature of every soft evidence function $w_i^z(z_0^i)$. See Appendix F for more details.

## 7 RELATED WORK AND CONCLUSION

Existing approaches for image inpainting can be divided into two classes: supervised methods and unsupervised methods. Specifically, supervised approaches require the model to be explicitly trained on inpainting tasks, while unsupervised approaches do not require task-specific training. Supervised methods are widely used for Variational Autoencoders (Peng et al., 2021; Zheng et al., 2019; Guo et al., 2019), Generative Adversarial Networks (Iizuka et al., 2017; Zhao et al., 2020; Guo et al., 2019), and Transformers (Yu et al., 2021; Wan et al., 2021). A major problem of supervised inpainting algorithms is that they are highly biased toward mask types observed during training and often need to be fine-tuned individually for every inpainting task (Xiang et al., 2023). However, since approximating the probability of masked pixels given known pixels is highly intractable for these models, we are unfortunately restricted to supervised inpainting approaches.

Recent developments in diffusion models (Ho et al., 2020; Song et al., 2020) open up the possibility for unsupervised inpainting as these models provide potential ways to approximate the constrained posterior. Specifically, the existence of variables at different noise levels allows us to blend in information from the known pixels to the denoising process of the diffusion models (Song & Ermon, 2019; Avrahami et al., 2022; Kawar et al., 2022). Additionally, techniques such as resampling images at higher noise levels (Lugmayr et al., 2022) and using partial filtering to approximate the constrained posterior (Trippe et al., 2022) greatly contribute to generating high-quality images.

The transition from supervised to unsupervised inpainting methods is a clear example that demonstrates the benefits of using more *tractable* models. Based on this observation, this paper seeks to further exploit tractable models. Specifically, we study the "extreme case", where we use TPMs that can *exactly* compute the constrained posterior. Empirical results suggest that TPMs can effectively improve the quality of inpainted images with only $10\%$ additional computational overhead.

**Acknowledgements**   This work was funded in part by the DARPA PTG Program under award HR00112220005, the DARPA ANSR program under award FA8750-23-2-0004, NSF grants #IIS-1943641, #IIS-1956441, #CCF-1837129, and a gift from RelationalAI. GVdB discloses a financial interest in RelationalAI. Mathias Niepert acknowledges funding by Deutsche Forschungsgemeinschaft (DFG, German Research Foundation) under Germany's Excellence Strategy - EXC and support by the Stuttgart Center for Simulation Science (SimTech).

**Reproducibility statement**   To facilitate reproducibility, we have clearly stated the proposed PC inference algorithm in Section 4.2. Algorithmic details including the choice of hyperparameters are provided in Appx. B and C.2. Formal proof of Thm. 1 is provided in Appendix A. We provide the code to train the PCs and to generate inpainted images at `https://github.com/UCLA-StarAI/Tiramisu`.

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

# Supplementary Material

## A   Proof of Theorem 1

The proof contains two main pairs: (i) shows that the forward pass computes $\sum_{\boldsymbol{x}_0} \prod_i w_i(\boldsymbol{x}_0^i) \cdot p(\boldsymbol{x}_0)$, and (ii) demonstrates the backward pass computes the conditional probabilities $p_{\text{TPM}}(\tilde{x}_0^i|\boldsymbol{x}_t, \boldsymbol{x}_0^{\text{k}})$.

**Correctness of the forward pass**   We show by induction that the forward value $\text{fw}_n$ of every node $n$ computes $\sum_{\boldsymbol{x}_0} \prod_{i \in \phi(n)} w_i(x_0^i) \cdot p(\boldsymbol{x}_0)$.

● Base case: input nodes. By definition, for every input node defined on variable $X_0^i := \phi(n)$, its forward value is $\sum_{\boldsymbol{x}_0} w_i(x_0^i) \cdot p_n(\boldsymbol{x}_0)$.

● Inductive case: product nodes. For every product node $n$, assume the forward value of its every child node $c$ satisfies $\text{fw}_c = \sum_{\boldsymbol{x}_0} \prod_{i \in \phi(c)} w_i(x_0^i) \cdot p_m(\boldsymbol{x}_0)$. Note that the forward value of product nodes is computed according to Equation (6), we have

$$\text{fw}_n = \prod_{c \in \text{in}(n)} \text{fw}_c,$$

$$= \prod_{c \in \text{in}(n)} \sum_{\boldsymbol{x}_0} \prod_{i \in \phi(c)} w_i(x_0^i) \cdot p_c(\boldsymbol{x}_0),$$

$$\overset{(a)}{=} \sum_{\boldsymbol{x}_0} \prod_{c \in \text{in}(n)} \prod_{i \in \phi(c)} w_i(x_0^i) \cdot p_c(\boldsymbol{x}_0),$$

$$\overset{(b)}{=} \sum_{\boldsymbol{x}_0} \prod_{i \in \phi(n)} w_i(x_0^i) \prod_{c \in \text{in}(n)} p_c(\boldsymbol{x}_0),$$

$$\overset{(c)}{=} \sum_{\boldsymbol{x}_0} \prod_{i \in \phi(n)} w_i(x_0^i) \cdot p_n(\boldsymbol{x}_0),$$

where $(a)$ follows from the fact that scopes of the children $\phi(c)$ are disjoint, and the child PCs $\{p_c(\boldsymbol{x}_0)\}_c$ are defined on disjoint sets; $(b)$ follows from the fact that $\phi(n) = \bigcup_{c \in \text{in}(n)} \phi(c)$; $(c)$ follows the definition in Equation (6).

● Inductive case: sum nodes. Similar to the case of product nodes, for every sum node $n$, we assume the forward value of its children satisfies the induction condition. We have

$$\text{fw}_n = \sum_{c \in \text{in}(n)} \theta_{n,c} \cdot \text{fw}_c,$$

$$= \sum_{c \in \text{in}(n)} \theta_{n,c} \sum_{\boldsymbol{x}_0} \prod_{i \in \phi(c)} w_i(x_0^i) \cdot p_c(\boldsymbol{x}_0),$$

$$\overset{(a)}{=} \sum_{\boldsymbol{x}_0} \prod_{i \in \phi(n)} w_i(x_0^i) \cdot \sum_{c \in \text{in}(n)} \phi_{n,c} \cdot p_c(\boldsymbol{x}_0),$$

$$\overset{(b)}{=} \sum_{\boldsymbol{x}_0} \prod_{i \in \phi(n)} w_i(x_0^i) \cdot p_n(\boldsymbol{x}_0),$$

where $(a)$ holds since $\forall c \in \text{in}(n), \phi(c) = \phi(n)$, and $(b)$ follows from Equation (6).

Therefore, the forward value of every node $n$ is defined by $\text{fw}_n = \sum_{\boldsymbol{x}_0} \prod_{i \in \phi(n)} w_i(x_0^i) \cdot p_m(\boldsymbol{x}_0)$.

**Correctness of the backward pass**   We first provide an intuitive semantics for the backward value $\text{bk}_n$ of every node: for every node $n$, if its forward value $\text{fw}_n$ were to set to $\text{fw}_n'$ (the other inputs of the PC remains unchanged), the value at the root node $n_r$ would change to $(1 - \text{bk}_n) \cdot \text{fw}_{n_r} + \text{bk}_n \cdot \text{fw}_{n_r} \cdot \text{fw}_n'/\text{fw}_n$. In the following, we prove this result by induction over the root node.

● Base case: the PC rooted at $n$. Denote $\text{bk}_n^{n_r}$ as the backward value of node $n$ w.r.t. the PC rooted at $n_r$. Since by definition $\text{bk}_n^n = 1$, we have that when the value of $n$ is changed to $\text{fw}_n'$, the root node's

value becomes

$$(1 - \mathrm{bk}_n^n) \cdot \mathrm{fw}_{n_r} + \mathrm{bk}_n^n \cdot \mathrm{fw}_n \cdot \mathrm{fw}_n' / \mathrm{fw}_n = \mathrm{fw}_n'.$$

• Inductive case: sum node. Suppose the statement holds for all children of a sum node $m$. Define $\mathrm{bk}_{n,c}^m$ as the backward value of edge $(n,c)$ for the PC rooted at $m$. Following the definition of the backward values, we have $\sum_{c \in \mathsf{in}(m)} \mathrm{bk}_{n,c}^m = \mathrm{bk}_n^m$. When the value of $n$ is changed to $\mathrm{fw}_n'$, the value of $m$ becomes:

$$\sum_{c \in \mathsf{in}(m)} \theta_{m,c} \cdot \mathrm{fw}_c' = \sum_{c \in \mathsf{in}(m)} \theta_{m,c} \cdot \big[ (1 - \mathrm{bk}_n^c) \cdot \mathrm{fw}_c + \mathrm{bk}_n^c \cdot \mathrm{fw}_c \cdot \mathrm{fw}_n' / \mathrm{fw}_n \big],$$

$$= \underbrace{\sum_{c \in \mathsf{in}(m)} \theta_{m,c} \cdot \mathrm{fw}_c}_{\mathrm{fw}_m} + \sum_{c \in \mathsf{in}(m)} \theta_{m,c} \cdot \mathrm{bk}_n^c \cdot \mathrm{fw}_c \cdot \big( \mathrm{fw}_n' / \mathrm{fw}_n - 1 \big),$$

$$= \mathrm{fw}_m + \sum_{c \in \mathsf{in}(m)} \theta_{m,c} \cdot \underbrace{\Big( \mathrm{bk}_{n,c}^m \cdot \frac{\mathrm{fw}_m}{\theta_{m,c} \cdot \mathrm{fw}_c} \Big)}_{\mathrm{bk}_n^c} \cdot \mathrm{fw}_c \cdot \big( \mathrm{fw}_n' / \mathrm{fw}_n - 1 \big),$$

$$= \mathrm{fw}_m + \underbrace{\sum_{c \in \mathsf{in}(m)} \mathrm{bk}_{n,c}^m}_{\mathrm{bk}_n^m} \cdot \mathrm{fw}_m \cdot \big( \mathrm{fw}_n' / \mathrm{fw}_n - 1 \big),$$

$$= (1 - \mathrm{bk}_n^m) \cdot \mathrm{fw}_m + \mathrm{bk}_n^m \cdot \mathrm{fw}_m \cdot \mathrm{fw}_n' / \mathrm{fw}_n,$$

• Inductive case: product node. Suppose the statement holds for all children of a product node $m$. Thanks to decomposability, at most one of $m$'s children can be an ancestor of $n$. Denote this child as $c'$. When the value of $n$ is changed to $\mathrm{fw}_n'$, the value of $m$ becomes:

$$\prod_{c \in \mathsf{in}(m), c \neq c'} \mathrm{fw}_c' = \prod_{c \in \mathsf{in}(m), c \neq c'} \mathrm{fw}_c \cdot \big[ (1 - \mathrm{bk}_n^{c'}) \cdot \mathrm{fw}_{c'} + \mathrm{bk}_n^{c'} \cdot \mathrm{fw}_{c'} \cdot \mathrm{fw}_n' / \mathrm{fw}_n \big],$$

$$\overset{(a)}{=} \prod_{c \in \mathsf{in}(m), c \neq c'} \mathrm{fw}_c \cdot \big[ (1 - \mathrm{bk}_n^m) \cdot \mathrm{fw}_{c'} + \mathrm{bk}_n^m \cdot \mathrm{fw}_{c'} \cdot \mathrm{fw}_n' / \mathrm{fw}_n \big],$$

$$\overset{(b)}{=} (1 - \mathrm{bk}_n^m) \cdot \mathrm{fw}_{c'} + \mathrm{bk}_n^m \cdot \mathrm{fw}_{c'} \cdot \mathrm{fw}_n' / \mathrm{fw}_n,$$

where $(a)$ holds since $\mathrm{bk}_n^m = \mathrm{bk}_n^{c'}$ and $(b)$ follows from the definition of product nodes in Equation (6).

Next, assume that the input nodes are all indicators in the form of $\mathbb{1}[X_i = x_i]$. In fact, any discrete univariate distribution can be represented as a mixture (sum node) of indicators. By induction, we can show that the sum of backward values of all input nodes corresponding to a variable $X_i$ is 1, since sum nodes only "divide" the backward value, and product nodes preserve the backward value sent to them. By setting the value of the input node $\mathbb{1}[X_i = x_i]$, we are essentially computing $\prod_{j \neq i} w_j(x_j) \cdot \mathbb{1}[X_i = x_i] \cdot p(\boldsymbol{x})$. Therefore, the backward values of these input nodes are proportional to the target conditional probability $p_{\mathrm{TPM}}(\tilde{x}_0^i | \boldsymbol{x}_t, \boldsymbol{x}_0^{\mathrm{k}})$.

We are left with proving that the sum of backward values of all input nodes corresponding to variable $X_i$ equals 1. To see this, consider the subset of nodes whose scope contains $X_i$. This subset of nodes represents a DAG with the root node as the only source node and input nodes of variables $X_i$ as the sink. Consider the backward algorithm as computing flows in the DAG. For every node non-sink node, the amount of flow it accepts equals the amount it sends. Specifically, product nodes send all their flow to their only child node (according to decomposability, at most one child of a product node contains $X_i$ in its scope); for every sum node $n$, the sum of flows sent to its children equal to the flow it receives. Since the flow sent by all source nodes equals the flow received by all sink nodes, we conclude that the backward values of the input nodes for variable $X_i$ sum up to 1. □

## B    Design Choices for High-Resolution Guided Image Inpainting

In all experiments, we compute $w_i^z(\tilde{z}_0^i)$ by first drawing 4 samples from $\frac{1}{Z} \prod_j w_j(\tilde{x}_0^j)$, and then feed them to the VQ-GAN's encoder. For every sample, we get a distribution over variable $\tilde{Z}_0^i$. $w_i^z(\tilde{z}_0^i)$ is then computed as the mean of the four distributions. In the following decoding phase that computes $p_{\text{TPM}}(\tilde{x}_0|\boldsymbol{x}_t, \boldsymbol{x}_0^{\text{k}}) := \mathbb{E}_{\tilde{\boldsymbol{z}}_0 \sim p_{\text{TPM}}(\cdot|\boldsymbol{x}_t, \boldsymbol{x}_0^{\text{k}})}[p(\tilde{\boldsymbol{x}}_0|\tilde{\boldsymbol{z}}_0)]$, we draw 8 random samples from $p_{\text{TPM}}(\cdot|\boldsymbol{x}_t, \boldsymbol{x}_0^{\text{k}})$ to estimate $p_{\text{TPM}}(\tilde{x}_0|\boldsymbol{x}_t, \boldsymbol{x}_0^{\text{k}})$.

In the following, we describe the hyperparameters of the adopted VQ-GAN for all three datasets:

Table 2: Hyperparameters of the adopted VQ-GAN models for Tiramisu.

|  | CelebA-HQ | ImageNet | LSUN-Bedroom |
| --- | --- | --- | --- |
| # latent variables | $16 \times 16$ | $16 \times 16$ | $16 \times 16$ |
| Vocab size | 1024 | 16384 | 16384 |

**Additional hyperparameters of Tiramisu**    For the distribution mixing hyperparameter $\alpha$ (cf. Section 3), we use an exponential decay schedule from $a$ to $b$ with a temperature parameter $\lambda$. Specifically, the mixing hyperparameter at time step $T$ is

$$(b - a) \cdot \exp(-\lambda \cdot t/T) + a. \tag{9}$$

We further define a cutoff parameter $t_{\text{cut}}$ such that when $t \leq t_{\text{cut}}$, the TPM guidance is not used. In all experiments, we used the first three samples in the validation set to tune the mixing hyperparameters. Hyperparameters are given in the following table. In all settings, we have $T = 250$.

Table 3: Mixing hyperparameters of Tiramisu.

|  | CelebA-HQ | ImageNet | LSUN-Bedroom |
| --- | --- | --- | --- |
| a | 0.8 | 0.8 | 0.8 |
| b | 1.0 | 1.0 | 1.0 |
| $\lambda$ | 2.0 | 2.0 | 2.0 |
| $t_{\text{cut}}$ | 200 | 235 | 235 |

## C    PC Learning Details

### C.1    The EM Parameter Learning Algorithm

As illustrated by Definition 1, when performing a feedforward evaluation (i.e., Equation (6)), a PC takes as input a sample $\boldsymbol{x}$ and outputs its probability $p_n(\boldsymbol{x})$. Given a dataset $\mathcal{D}$, our goal is to learn a set of PC parameters (including sum edge parameters and input node/distribution parameters) to maximize the MLE objective: $\sum_{\boldsymbol{x} \in \mathcal{D}} \log p_n(\boldsymbol{x})$. The Expectation-Maximization algorithm is a natural way to learn PC parameters since PCs can be viewed as latent variable models with a hierarchically nested latent space (Peharz et al., 2016). There are two interpretations of the EM learning algorithm for PCs: one based on gradients (Peharz et al., 2020) and the other based on a new concept called circuit flows (Choi et al., 2021). We use the gradient-based interpretation since it is easier to understand.

Note that the feedforward computation of PCs is differentiable and can be modeled by a computation graph. Therefore, after computing the log-likelihood $\log p_n(\boldsymbol{x})$ of a sample $\boldsymbol{x}$, we can efficiently compute its gradient with respect to all PC parameters via the backpropagation algorithm. Given a mini-batch of samples, we use the backpropagation algorithm to accumulate gradients for every parameter. Take sum parameters as an example. Define the cumulative gradient of $\theta_{n,c}$ as $g_{n,c}$, the updated parameters $\{\theta_{n,c}\}_{c \in \text{in}(n)}$ for every sum node $n$ is given by:

$$\theta_{n,c} \leftarrow (1 - \alpha) \cdot \theta_{n,c} + \alpha \cdot \frac{g_{n,c} \cdot \theta_{n,c}}{\sum_{m \in \text{in}(n)} g_{n,m} \cdot \theta_{n,m}},$$

where $\alpha \in (0, 1]$ is the step size. For input nodes, since we only use categorical distributions that can be equivalently represented as a mixture over indicator leaves (i.e., a sum node connecting these indicators), optimizing leaf parameters is equivalent to the sum parameter learning process.

## C.2  DETAILS OF THE PC LEARNING PIPELINE

**PC structure**  We adopt a variant of the original PD structure proposed in Poon & Domingos (2011). Specifically, starting from the whole image, the PD structure gradually uses product nodes to horizontally or vertically split the variable scope into half. As a result, the scope of every node represents a patch (of variable size) in the original image. We use categorical input nodes in accordance with the VQ-GAN's latent space. We treat the set of parameters belonging to nodes with the same scope as the parameters of the scope. Based on the original structure, we further tie the parameters representing every pixel and every $2 \times 2$ patch. Since the marginal distribution of every latent variable should be similar thanks to the spatial invariance of images.

**Parameter learning**  This paper uses the latent variable distillation (LVD) technique (Liu et al., 2022b; 2023) to initiate the PC parameters before further fine-tuning them with the EM algorithm described in Appendix C.1. Intuitively, LVD provides extra supervision to PC optimizers through semantic-aware latent variable assignments extracted from deep generative models. We refer readers to the original papers for more details.

After initializing PC parameters with LVD, we further fine-tune the parameters with EM with the following hyperparameters:

Table 4: Hyperparameters of EM fine-tuning process.

| Name | Value |
| --- | --- |
| Step size | 1.0 |
| Batch size | 20000 |
| Pseudocount | 0.1 |
| # iterations | 200 |

## D  DETAILS OF THE MAIN EXPERIMENTS AND THE BASELINES

**Adopted Masks**  Six of the seven adopted mask types are shown in Figure 6 (gray indicates masked region). We adopt the wide masks from Lugmayr et al. (2022); Suvorov et al. (2022). There are 100 masks generated by uniformly sampling from polygonal chains dilated by a high random width and rectangles of arbitrary aspect ratios.

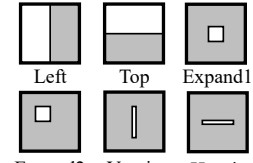

Figure 6: Used masks.

The "wide" masks can be downloaded from `https://drive.google.com/uc?id=1Q_dxuyI41AAmSv9ti3780BwaJQqwvwMv`, which is provided by (Lugmayr et al., 2022).

**Pretrained Models**  For all inpainting algorithms, we adopt the same diffusion model checkpoint pretrained by Lugmayr et al. (2022) (for CelebA-HQ) and OpenAI (for ImageNet and LSUN-Bedroom; `https://github.com/openai/guided-diffusion`). The links to the checkpoints for all three datasets are listed below.

- CelebA-HQ:   `https://drive.google.com/uc?id=1norNWWGYP3EZ_o05DmoW1ryKuKMmhlCX`

- ImageNet:   `https://openaipublic.blob.core.windows.net/diffusion/jul-2021/256x256_diffusion.pt`  and  `https://openaipublic.blob.core.windows.net/diffusion/jul-2021/256x256_classifier.pt`.

- LSUN-Bedroom: `https://openaipublic.blob.core.windows.net/diffusion/jul-2021/lsun_bedroom.pt`.

**Data**  For CelebA-HQ and LSUN-Bedroom, we use the first 100 images in the validation set. We adopt the validation split of CelebA-HQ following Suvorov et al. (2022). For ImageNet, we use a random validation image for the first 100 classes.

Table 5: User study results. We report the vote difference (%), i.e., [percentage of votes to Tiramisu] - [percentage of votes to the baseline]. The higher the vote difference value, the more the annotators prefer images generated by Tiramisu compared to the baseline.

| Tasks | | Algorithms | | | |
|---|---|---|---|---|---|
| Dataset | Mask | Tiramisu (ours) | CoPaint | RePaint | DPS |
| CelebA-HQ | Expand1 | Reference | 22 | 34 | 14 |
| | Wide | Reference | 26 | 30 | 22 |
| ImageNet | Expand1 | Reference | 32 | 20 | 24 |
| | Wide | Reference | 14 | 6 | 12 |
| LSUN-Bedroom | Expand1 | Reference | 18 | 2 | 8 |
| | Wide | Reference | 4 | 6 | -6 |

# E ADDITIONAL EXPERIMENTS

## E.1 USER STUDY

Since image inpainting is an ill-posed problem and LPIPS alone may not be sufficient to indicate the performance of each algorithm, we recruited human evaluators to evaluate the quality of inpainted images. Specifically, for every baseline method, we sample inpainted image pairs from the baseline and Tiramisu using the same inputs (source image and mask). For every image pair, the evaluator is provided with both inpainted images and is tasked to select the better one based on the following criterion: images that visually look more natural and without artifacts (e.g., blurry, distorted). A screenshot of the interface is shown in Figure 7.

The user study is conducted on the three strongest baselines based on their overall LPIPS scores: CoPaint (Zhang et al., 2023a), RePaint (Lugmayr et al., 2022), and DPS (Chung et al., 2022). We use two mask types for comparison: "expand1" and "wide". For every comparison, we report the vote difference (%), which is the percentage of votes to Tiramisu subtracted by that of the baseline. A positive vote difference value means images generated by Tiramisu are preferred compared to the baselines, while a negative value suggests that the baseline is better than Tiramisu.

We adopt the three most competitive baselines, i.e., CoPaint, RePaint, and DPS, based on their average LPIPS scores (Table 1). For all three datasets, we conduct user studies on two types of masks: "expand1" and "wide". Results are shown in Table 5. The vote difference scores are mostly high, indicating the superior inpainting performance of Tiramisu. Additionally, we observe that Tiramisu generally performs better with the "expand1" mask (with larger to-be-inpainted regions), which suggests that Tiramisu may be more helpful in the case of large-hole image inpainting.

## E.2 ADDITIONAL QUALITATIVE RESULTS

Please refer to Figure 8 to 10 for additional qualitative results on all three adopted datasets.

# F DETAILS OF THE SEMANTIC FUSION EXPERIMENT

The mixing hyperparameters are the same as described in Appendix B.

The VQ-GAN encoder first generates an embedding $\mathbf{e}$ for every latent variable $\tilde{Z}_0^i$, and it is then discretized with the VQ codebook by selecting the ID in the codebook that has the minimum L2 distance with $\mathbf{e}$. We soften this process by setting $w_i^z(j) = \exp(-\|\mathbf{e} - \mathbf{e}_j\|_2/\lambda_{\text{sf}})$, where $\mathbf{e}_j$ is the $j$th embedding in the codebook, and $\lambda_{\text{sf}}$ is the temperature that controls the semantic coherence level of the generated images. The closer $\lambda_{\text{sf}}$ is to 0, the higher the coherence level.

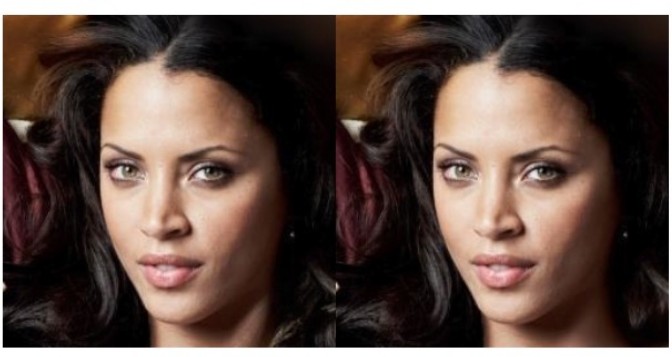

Figure 7: User study interface.

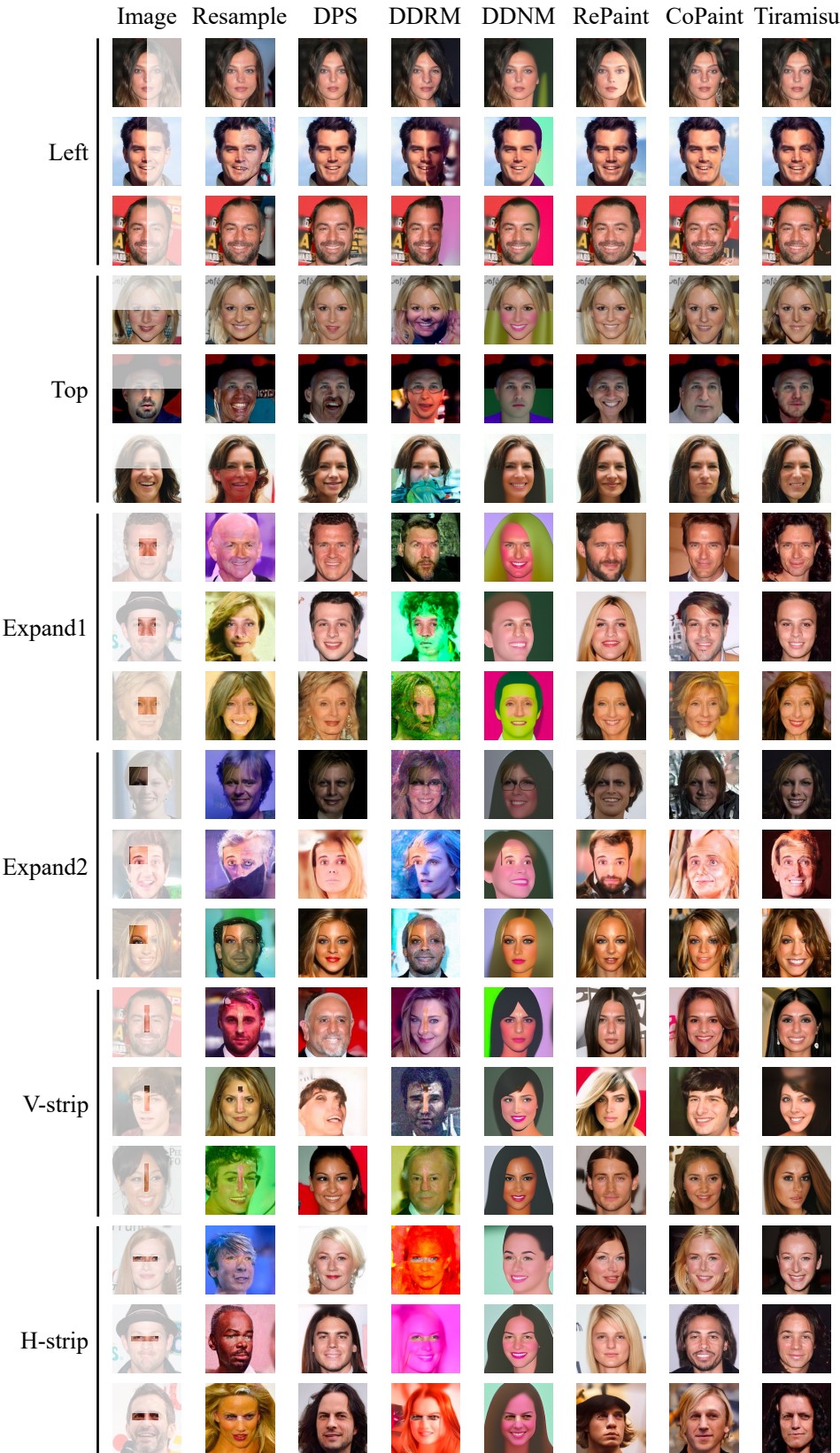

Figure 8: Additional qualitative results on CelebA-HQ with six mask types.

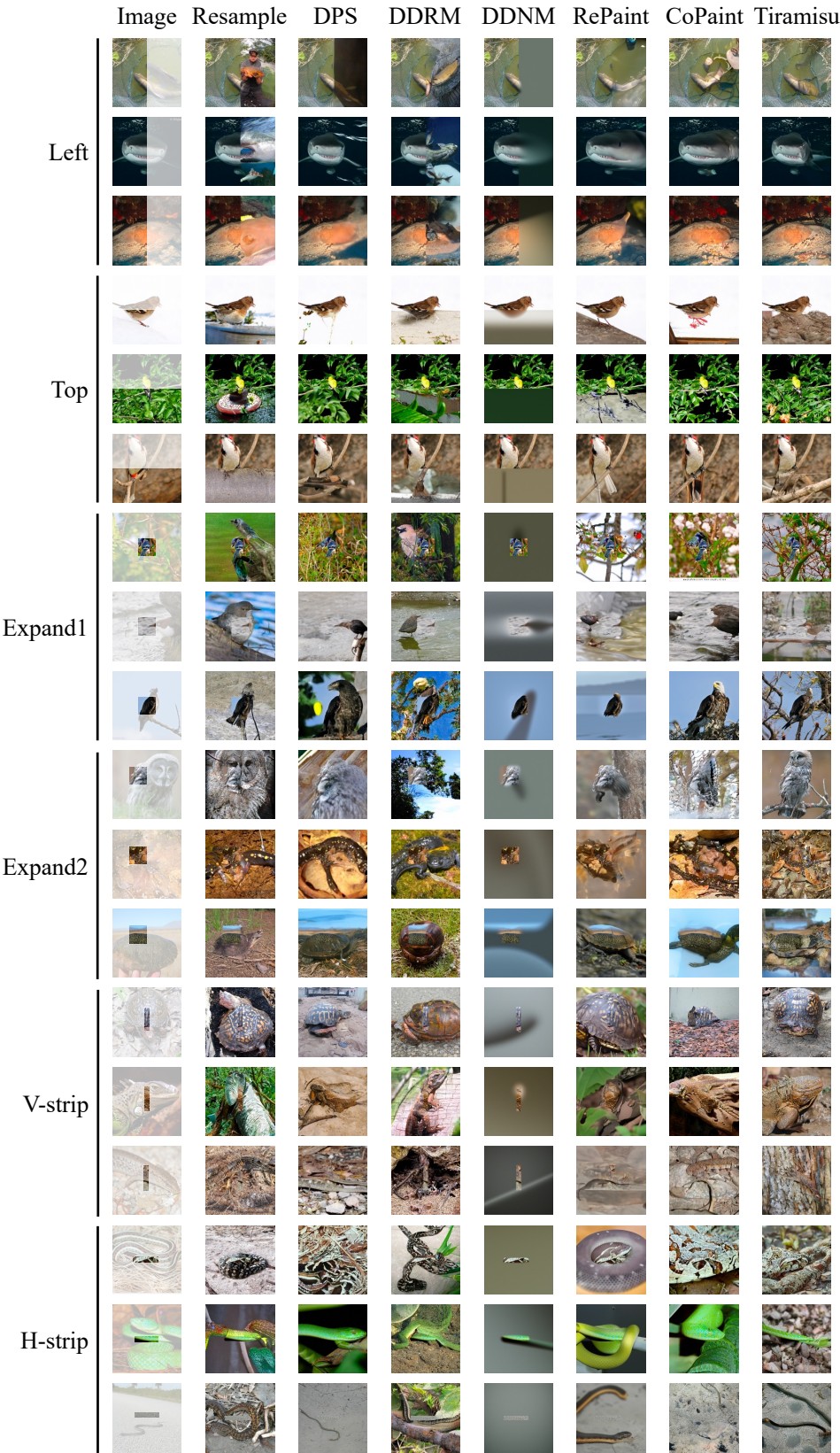

Figure 9: Additional qualitative results on ImageNet with six mask types.

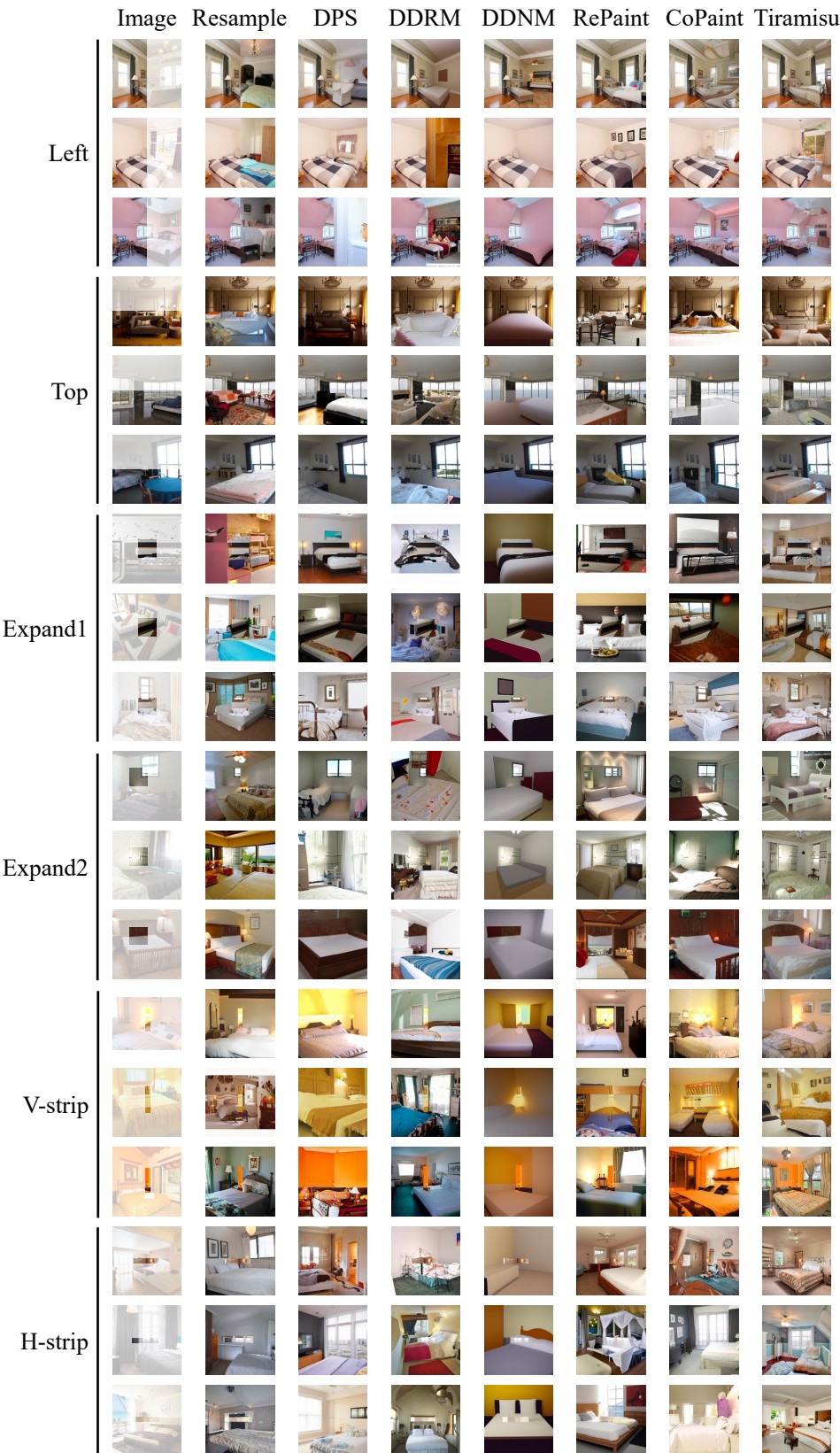

Figure 10: Additional qualitative results on LSUN-Bedroom with six mask types.

