# OpenReview forum: "Image Inpainting via Tractable Steering of Diffusion Models"
_ICLR.cc/2024/Conference — ICLR 2024 poster_

### Official Review · Reviewer_dUar · 2023-10-24

**Soundness:** 3 good
**Presentation:** 2 fair
**Contribution:** 3 good
**Rating:** 5
**Confidence:** 3

**Summary:**

This paper introduces a method of incorporating Probabilistic Circuits (PCs) into diffusion models, with the authors claiming that this approach can encourage diffusion models to generate structurally more coherent images in image inpainting.

**Strengths:**

In this paper, the Tractable Probabilistic Models (TPMs) are first introduced into the task of controllable image inpainting. Experimental results demonstrate that this approach encourages the model to generate higher-quality samples with only a limited computational cost increase.
The experimental results seem visually plausible.

**Weaknesses:**

1) The readability of this paper is relatively low. We believe that the author should explain in the introduction and method sections whether the method proposed is training or non-training. If it is the former, the loss function and the parts of the network that need to be updated should be stated explicitly. If it is the latter, the pseudo-code of the algorithm should be given.

2) The experimental section lacks metrics.

3) Have the authors tried irregular masks?

**Questions:**

1) I cannot fully understand the details of the method proposed in this paper (e.g., how were the weights in the PC obtained). I would appreciate it if the authors could provide pseudocode to enhance the method's readability further.

2) LPIPS alone may not be sufficient to assess the quality of generated images. We recommend that the authors report metrics such as FID, U-IDS, etc.


If I have misunderstood, please point it out.

I am very willing to improve the rating after reading your rebuttal and considering the opinions of other reviewers.

---

> ### Author Response · Authors · 2023-11-19
> **Response to Reviewer dUar**
>
> > We believe that the author should explain in the introduction and method sections whether the method proposed is training or non-training. … I cannot fully understand the details of the method proposed in this paper (e.g., how were the weights in the PC obtained)
>
> Thank you for pointing out aspects of the paper that were not explained clearly in the main text. The PC used in the proposed inpainting algorithm needs to be first trained (unconditionally) on the datasets (e.g., CelebA-HQ, ImageNet, and LSUN-Bedroom) using the MLE objective: $\max \sum_{\mathbf{x} \in \mathcal{D}} \log p_{n} (\mathbf{x})$, where $p_{n}$ is the PC and $\mathcal{D}$ is the dataset.
>
> Following prior art, a natural way to optimize PC parameters w.r.t. the MLE objective is by the Expectation-Maximization algorithm since PCs can be viewed as a latent variable model with a hierarchically nested latent space. The implementation of the EM algorithm is very similar to the backpropagation algorithm since the feedforward computation following Definition 1 can be treated as a computational graph. To update the parameters, we first run the standard backpropagation algorithm to compute the gradient of all parameters (w.r.t. the MLE objective, which is the output of the feedforward PC computation). Suppose the gradient of $\theta_{n,c}$ is $g_{n,c}$, for every sum node $n$, we update all its child parameters by
>
> $$\theta_{n,c} \leftarrow (1 - \alpha) \cdot \theta_{n,c} + \alpha \cdot \frac{g_{n,c} \cdot \theta_{n,c}}{\sum_{m \in \mathtt{in}(n)} g_{n,m} \cdot \theta_{n,m}},$$
>
> where $\alpha$ is the step size. In addition to the EM update, we follow prior arts and use the LVD technique to initialize the parameters. The full details of the learning process of PCs are included in Appendix C of the revised paper. At the end of Section 4.1, we also added a paragraph to explain the PC learning pipeline and points to the Appendix for more details.
>
> We are happy to make further clarifications if there are other unclear parts in the paper.
>
> > LPIPS alone may not be sufficient to assess the quality of generated images.
>
> We conduct a user study based on all three datasets and two masks (“Expand1” and the newly added “Wide” masks adopted from [1,2]). Detailed settings and results are added in the updated paper (Appendix E.1). We copy the result table from the paper:
>
> | &nbsp;&nbsp;&nbsp;&nbsp; Dataset & Mask     &nbsp;&nbsp; &nbsp;  |  &nbsp;  Tiramisu &nbsp; | CoPaint | RePaint | DPS |
>
> | CelebAHQ & Expand1  |  Reference    |   &nbsp;&nbsp;&nbsp;&nbsp;  22    &nbsp;&nbsp;&nbsp;&nbsp;  |    &nbsp;&nbsp;&nbsp;&nbsp; 34   &nbsp;&nbsp;&nbsp;&nbsp;  |  &nbsp;14 &nbsp; |
>
> | CelebAHQ & Wide  &nbsp; &nbsp; &nbsp; |  Reference    |   &nbsp;&nbsp;&nbsp;&nbsp;  26    &nbsp;&nbsp;&nbsp;&nbsp;  |    &nbsp;&nbsp;&nbsp;&nbsp; 30   &nbsp;&nbsp;&nbsp;&nbsp;  |  &nbsp; 22 &nbsp; |
>
> | ImageNet & Expand1  |  Reference    |   &nbsp;&nbsp;&nbsp;&nbsp;  32    &nbsp;&nbsp;&nbsp;&nbsp;  |    &nbsp;&nbsp;&nbsp;&nbsp; 20   &nbsp;&nbsp;&nbsp;&nbsp;  |  &nbsp; 24 &nbsp; |
>
> | ImageNet & Wide &nbsp;&nbsp; &nbsp;&nbsp;    |  Reference    |   &nbsp;&nbsp;&nbsp;&nbsp;  14    &nbsp;&nbsp;&nbsp;&nbsp;  |    &nbsp;&nbsp;&nbsp;&nbsp;&nbsp; 6   &nbsp;&nbsp;&nbsp;&nbsp;&nbsp;  |  &nbsp; 12 &nbsp; |
>
> | LSUN & Expand1&nbsp; &nbsp;&nbsp; &nbsp; &nbsp;    |  Reference    |   &nbsp;&nbsp;&nbsp;&nbsp;  18    &nbsp;&nbsp;&nbsp;&nbsp;  |    &nbsp;&nbsp;&nbsp;&nbsp;&nbsp; 2   &nbsp;&nbsp;&nbsp;&nbsp;&nbsp;  |  &nbsp; 8 &nbsp; |
>
> | LSUN & Wide &nbsp;&nbsp; &nbsp; &nbsp; &nbsp; &nbsp; &nbsp; &nbsp;   |  Reference    |   &nbsp;&nbsp;&nbsp;&nbsp; &nbsp; 4    &nbsp;&nbsp;&nbsp;&nbsp; &nbsp; |    &nbsp;&nbsp;&nbsp;&nbsp;&nbsp; 6   &nbsp;&nbsp;&nbsp;&nbsp;&nbsp;  |  &nbsp; -6 &nbsp; |
>
> We use the vote difference (%) metric, which is the percentage of votes to Tiramisu subtracted by that of the baseline. A positive vote difference value means images generated by Tiramisu are preferred compared to the baselines, while a negative value suggests that the baseline is better than Tiramisu.
>
> We compare against the three strongest baselines (CoPaint, RePaint, and DPS) as their average LPIPS scores are not significantly worse than Tiramisu. As shown in the table, in all but one task the vote difference score is positive (and often quite large), which indicates the superiority of Tiramisu compared to the baselines.
>
> We also tried the metrics suggested by the reviewer. However, FID is not specific to image pairs and is less informative for image inpainting tasks; when using the official U-IDS metric, we found that in most cases we got a U-IDS close to 0 and the results are not very informative.

---

> > ### Author Response · Authors · 2023-11-23
> >
> > Dear Reviewer dUar,
> >
> > We were wondering if you have had a chance to read our reply to your feedback. As the time window for the rebuttal is closing soon, please let us know if there are any additional questions we can answer.
> >
> > Best,
> >
> > Authors

---

### Official Review · Reviewer_rz93 · 2023-10-31

**Soundness:** 3 good
**Presentation:** 3 good
**Contribution:** 3 good
**Rating:** 6
**Confidence:** 5

**Summary:**

This paper investigated image inpainting using a pre-trained diffusion model by using the proposed TPM approach to enforce the inpainting constraints. Unlike repaint or copaint that either enforce the inpainting constraints through the time travel technique or gradient-based methods to optimize, this paper proposes to model p_\theta{x_0} by PC, which turns the computation into a normalized multiplication of the model weight based on the graph. The proposed method was combined with copaint and compared with existing methods. The LPIPS value got slightly improved on three commonly used datasets: CelebA-HQ, ImageNet, LSUN-Bedroom. Visual examples were presented and computing cost was discussed.

**Strengths:**

- Inpainting using pre-trained diffusion is an interesting direction to explore and modeling p_\theta{x_0} provides an alternative angle to look into this problem.
- The presentation is clear and results show the potential of the proposed method. On the three datasets, CelebA-HQ, ImageNet, LSUN-Bedroom, with masks of different shapes or at different positions, the proposed approach achieved higher LPIPS scores mostly.
- A practical approximation to apply the proposed methods for high-resolution imprinting was discussed and proposed.

**Weaknesses:**

- The computing cost is related to the size of the hole as well as the network architecture. As authors have mentioned, it also related to resolution. So instead of claiming 10% additional computation overhead, a detailed analysis is more helpful.
- Number-wise the improvement on LPIPS value compared to CoPaint, or even RePaint is minor. How about other metrics? Or user studies?
- Section 4.2 reads slightly disconnected from Section 4.1, especially the introduction of the equation 7.

**Questions:**

Will code be released upon publication of this work?

---

> ### Author Response · Authors · 2023-11-19
> **Response to Reviewer rz93 (part 1)**
>
> We thank the reviewer for their constructive feedback.
>
> > So instead of claiming 10% additional computation overhead, a detailed analysis is more helpful.
>
> Thank you for the suggestion. In the updated paper (Section 6.2, the “computational efficiency” paragraph), we provide an ablation study over the % of denoising steps using PCs, and report the LPIPS score and runtime-per-image for each case. We adopt the “CelebA-HQ + the Expand1 mask” task. As shown in Figure 4 of the updated paper (for convenience, we copied the results in the figure to the table below) as we engage the PC in more denoising steps, the LPIPS score first decreases and then increases. Therefore, incorporating PCs in a moderate amount of steps gives the best performance (around 20% in this case). In terms of computational efficiency, the additional computation cost is around 10% when we engage the PC for the first 20% of the denoising steps and around 25% when the PC is used in the first 50% of the denoising steps.
>
> One explanation for the diminishing performance gain when using PCs in too many denoising steps is that the later stages of denoising are mainly responsible for refining the details in the image. And the guidance of PCs could be less effective in this aspect. This could be the consequence of using latent space PC (latent space generated by VQ-GAN). In the future, we will explore possibilities to train good PCs directly in the pixel space. This could lead to better performance in terms of refining details in generated images.
>
> | % denoising steps using PC  |   &nbsp;&nbsp;&nbsp;  5    &nbsp;&nbsp;&nbsp;&nbsp;|&nbsp;&nbsp;&nbsp;   10    &nbsp;&nbsp;&nbsp;|&nbsp; &nbsp;&nbsp;   20   &nbsp;&nbsp;&nbsp;|&nbsp;&nbsp;&nbsp;    30   &nbsp;&nbsp;&nbsp;|&nbsp;&nbsp;&nbsp;   40    &nbsp;&nbsp;&nbsp;|&nbsp;&nbsp;&nbsp;  50 &nbsp;&nbsp; |
>
> | &nbsp;&nbsp;&nbsp;&nbsp;&nbsp;&nbsp;&nbsp;&nbsp;&nbsp;&nbsp;&nbsp;&nbsp;&nbsp;&nbsp;&nbsp;&nbsp;&nbsp;&nbsp;&nbsp; LPIPS &nbsp;&nbsp;&nbsp;&nbsp;&nbsp;&nbsp;&nbsp;&nbsp;&nbsp;&nbsp;&nbsp;&nbsp;&nbsp;&nbsp;&nbsp;&nbsp;&nbsp;&nbsp;&nbsp; |  0.470 | 0.459 | &nbsp; 0.454  | 0.456 | 0.464 &nbsp; | 0.476 |
>
> | &nbsp;&nbsp;&nbsp;&nbsp;&nbsp;&nbsp;&nbsp;&nbsp;&nbsp;&nbsp;&nbsp;&nbsp;&nbsp;&nbsp;&nbsp;&nbsp; Runtime &nbsp;&nbsp;&nbsp;&nbsp;&nbsp;&nbsp;&nbsp;&nbsp;&nbsp;&nbsp;&nbsp;&nbsp;&nbsp;&nbsp;&nbsp;&nbsp; |  &nbsp; 109 &nbsp;  | &nbsp; 111 &nbsp;  |  &nbsp; 115 &nbsp;  |  &nbsp; 118 &nbsp; |  &nbsp;121 &nbsp;  | &nbsp; 125 &nbsp; |

---

> > ### Author Response · Authors · 2023-11-19
> > **Response to Reviewer rz93 (part 2)**
> >
> > > How about other metrics? Or user studies?
> >
> > We conduct a user study based on all three datasets and two masks (“Expand1” and the newly added “Wide” masks adopted from [1,2]). Detailed settings and results are added in the updated paper (Appendix E.1). We copy the result table from the paper:
> >
> > | &nbsp;&nbsp;&nbsp;&nbsp; Dataset & Mask     &nbsp;&nbsp; &nbsp;  |  &nbsp;  Tiramisu &nbsp; | CoPaint | RePaint | DPS |
> >
> > | CelebAHQ & Expand1  |  Reference    |   &nbsp;&nbsp;&nbsp;&nbsp;  22    &nbsp;&nbsp;&nbsp;&nbsp;  |    &nbsp;&nbsp;&nbsp;&nbsp; 34   &nbsp;&nbsp;&nbsp;&nbsp;  |  &nbsp;14 &nbsp; |
> >
> > | CelebAHQ & Wide  &nbsp; &nbsp; &nbsp; |  Reference    |   &nbsp;&nbsp;&nbsp;&nbsp;  26    &nbsp;&nbsp;&nbsp;&nbsp;  |    &nbsp;&nbsp;&nbsp;&nbsp; 30   &nbsp;&nbsp;&nbsp;&nbsp;  |  &nbsp; 22 &nbsp; |
> >
> > | ImageNet & Expand1  |  Reference    |   &nbsp;&nbsp;&nbsp;&nbsp;  32    &nbsp;&nbsp;&nbsp;&nbsp;  |    &nbsp;&nbsp;&nbsp;&nbsp; 20   &nbsp;&nbsp;&nbsp;&nbsp;  |  &nbsp; 24 &nbsp; |
> >
> > | ImageNet & Wide &nbsp;&nbsp; &nbsp;&nbsp;    |  Reference    |   &nbsp;&nbsp;&nbsp;&nbsp;  14    &nbsp;&nbsp;&nbsp;&nbsp;  |    &nbsp;&nbsp;&nbsp;&nbsp;&nbsp; 6   &nbsp;&nbsp;&nbsp;&nbsp;&nbsp;  |  &nbsp; 12 &nbsp; |
> >
> > | LSUN & Expand1&nbsp; &nbsp;&nbsp; &nbsp; &nbsp;    |  Reference    |   &nbsp;&nbsp;&nbsp;&nbsp;  18    &nbsp;&nbsp;&nbsp;&nbsp;  |    &nbsp;&nbsp;&nbsp;&nbsp;&nbsp; 2   &nbsp;&nbsp;&nbsp;&nbsp;&nbsp;  |  &nbsp; 8 &nbsp; |
> >
> > | LSUN & Wide &nbsp;&nbsp; &nbsp; &nbsp; &nbsp; &nbsp; &nbsp; &nbsp;   |  Reference    |   &nbsp;&nbsp;&nbsp;&nbsp; &nbsp; 4    &nbsp;&nbsp;&nbsp;&nbsp; &nbsp; |    &nbsp;&nbsp;&nbsp;&nbsp;&nbsp; 6   &nbsp;&nbsp;&nbsp;&nbsp;&nbsp;  |  &nbsp; -6 &nbsp; |
> >
> > We use the vote difference (%) metric, which is the percentage of votes to Tiramisu subtracted by that of the baseline. A positive vote difference value means images generated by Tiramisu are preferred compared to the baselines, while a negative value suggests that the baseline is better than Tiramisu.
> >
> > We compare against the three strongest baselines (CoPaint, RePaint, and DPS) as their average LPIPS scores are not significantly worse than Tiramisu. As shown in the table, in all but one task the vote difference score is positive (and often quite large), which indicates the superiority of Tiramisu compared to the baselines.
> >
> >
> > > Section 4.2 reads slightly disconnected from Section 4.1, especially the introduction of the equation 7.
> >
> > Thank you for spotting this. Section 4.2 continues to introduce how to compute Equation (4) (from the TPM), which is a key step in the general algorithm described in Section 3. In the revised manuscript, we have added clarification sentences at the beginning of Section 4.2 to remind readers why we need to compute $p_{TPM} (\tilde{\mathbf{x}}_0 | \mathbf{x}_t, \mathbf{x}_0^k)$.
> >
> > > Will code be released upon publication of this work?
> >
> > Yes, we will release the training/inference code as well as the pretrained PCs if this work gets accepted.

---

### Official Review · Reviewer_fJmV · 2023-10-31

**Soundness:** 2 fair
**Presentation:** 2 fair
**Contribution:** 2 fair
**Rating:** 6
**Confidence:** 3

**Summary:**

This paper introduces a novel method that integrates Tractable Probabilistic Models, particularly Probabilistic Circuits, to address the challenges in controlling diffusion models for image inpainting tasks. This approach aims to achieve more precise and efficient inpainting by leveraging the exact computation of constrained posteriors provided by TPMs.​

**Strengths:**

The paper appears to present a novel approach to the problem of image inpainting using diffusion models. The integration of Tractable Probabilistic Models (TPMs), specifically Probabilistic Circuits (PCs), to guide the denoising process of diffusion models is an inventive combination of existing ideas.
This creative synergy seems to address the intractability issue inherent in exact conditioning required for tasks like inpainting. Additionally, the paper builds upon prior advances to scale up PCs for guiding the image generation process, which could be considered an original contribution in terms of improving and extending existing methodologies.

**Weaknesses:**

1. The TPMs seem to be a general design, while this work constrain the application to image inpainting only, I am not sure about the intuition of this specific application. How about the potential of this method for general conditional generation?

2. In section 6.2, it says "we only need to incorporate guidance from the TPM in the early denoising stages to control the global semantics of the image; fine-grained details can be later refined by the diffusion model. As a result, TPM is only required in the first ∼20% denoising steps". Here, more experimental analysis of the TPM steps are expected, including the effect on generation quality, semantic coherence, and computational efficiency.

3. Limitations and Failure Cases: A discussion of the method's limitations and failure cases are not addressed in this research, which would be beneficial for the application of this work.

**Questions:**

See above.

---

> ### Author Response · Authors · 2023-11-19
> **Response to Reviewer fJmV**
>
> We thank the reviewer for their constructive feedback.
>
> > How about the potential of this method for general conditional generation?
>
> There are various conditional/constrained image generation tasks that could be done by (variants of) the proposed method. As a first attempt at using TPMs (PCs) for conditional image generation tasks, this paper uses image inpainting to demonstrate the effectiveness and potential of the proposed method. In the following, we list several classes of conditional generation tasks that could be accomplished by the proposed method or some (non-trivial) extension of our method.
>
> - Other types of constraints belong to the independent soft evidence constraint family (formally defined in Section 4.2). For example, the constraint of image super-resolution and deblurring could be written in the form of such constraints and thus can be tackled by the proposed method.
>
> - Other inverse problems that can be formulated into “simple” constraints could potentially be tackled by a generalized version of the proposed method thanks to the ability of PCs to efficiently *condition on certain logical constraints*. For example, in image coloring, the constraint would be that for every pixel, its R, G, B value have a fixed weighted sum (specified by the input gray-scale image).
>
> - Controlled generation tasks without formally described conditions. For example, the semantic fusion task described in Section 6.3 aims to “fuse” the semantics of several reference images while maintaining the “naturalness” of the resultant image.
>
> > Here, more experimental analysis of the TPM steps are expected, including the effect on generation quality, semantic coherence, and computational efficiency.
>
> Thank you for the suggestion. In the updated paper (Section 6.2, the “computational efficiency” paragraph), we provide an ablation study over the % of denoising steps using PCs, and report the LPIPS score and runtime-per-image for each case. We adopt the “CelebA-HQ + the Expand1 mask” task. As shown in Figure 4 of the updated paper (for convenience, we copied the results in the figure to the table below) as we engage the PC in more denoising steps, the LPIPS score first decreases and then increases. Therefore, incorporating PCs in a moderate amount of steps gives the best performance (around 20% in this case). In terms of computational efficiency, the additional computation cost is around 10% when we engage the PC for the first 20% denoising steps and around 25% when the PC is used in the first 50% denoising steps.
>
> One explanation for the diminishing performance gain when using PCs in too many denoising steps is that the later stages of denoising are mainly responsible for refining the details in the image. And the guidance of PCs could be less effective in this aspect. This could be the consequence of using latent space PC (latent space generated by VQ-GAN). In the future, we will explore possibilities to train good PCs directly in the pixel space. This could lead to better performance in terms of refining details in generated images.
>
> | % denoising steps using PC  |   &nbsp;&nbsp;&nbsp;  5    &nbsp;&nbsp;&nbsp;&nbsp;|&nbsp;&nbsp;&nbsp;   10    &nbsp;&nbsp;&nbsp;|&nbsp; &nbsp;&nbsp;   20   &nbsp;&nbsp;&nbsp;|&nbsp;&nbsp;&nbsp;    30   &nbsp;&nbsp;&nbsp;|&nbsp;&nbsp;&nbsp;   40    &nbsp;&nbsp;&nbsp;|&nbsp;&nbsp;&nbsp;  50 &nbsp;&nbsp; |
>
> | &nbsp;&nbsp;&nbsp;&nbsp;&nbsp;&nbsp;&nbsp;&nbsp;&nbsp;&nbsp;&nbsp;&nbsp;&nbsp;&nbsp;&nbsp;&nbsp;&nbsp;&nbsp;&nbsp; LPIPS &nbsp;&nbsp;&nbsp;&nbsp;&nbsp;&nbsp;&nbsp;&nbsp;&nbsp;&nbsp;&nbsp;&nbsp;&nbsp;&nbsp;&nbsp;&nbsp;&nbsp;&nbsp;&nbsp; |  0.470 | 0.459 | &nbsp; 0.454  | 0.456 | 0.464 &nbsp; | 0.476 |
>
> | &nbsp;&nbsp;&nbsp;&nbsp;&nbsp;&nbsp;&nbsp;&nbsp;&nbsp;&nbsp;&nbsp;&nbsp;&nbsp;&nbsp;&nbsp;&nbsp; Runtime &nbsp;&nbsp;&nbsp;&nbsp;&nbsp;&nbsp;&nbsp;&nbsp;&nbsp;&nbsp;&nbsp;&nbsp;&nbsp;&nbsp;&nbsp;&nbsp; |  &nbsp; 109 &nbsp;  | &nbsp; 111 &nbsp;  |  &nbsp; 115 &nbsp;  |  &nbsp; 118 &nbsp; |  &nbsp;121 &nbsp;  | &nbsp; 125 &nbsp; |
>
>
> > Limitations and Failure Cases: A discussion of the method's limitations and failure cases are not addressed in this research, which would be beneficial for the application of this work.
>
> As described in the response to the above question, engaging the guidance of PCs in too many denoising steps could lead to worse performance. This suggests that the current method may not be good at refining details. Therefore, Tiramisu could be less effective when dealing with small-hole inpainting.
>
> Another potential weakness of the proposed method is the expressiveness of PCs. While we are able to train good PCs on datasets such as ImageNet 256*256, it is not sure whether this could be generalized to higher-dimensional images.

---

### Official Review · Reviewer_yw5k · 2023-11-01

**Soundness:** 3 good
**Presentation:** 3 good
**Contribution:** 3 good
**Rating:** 5
**Confidence:** 3

**Summary:**

This paper proposes to use TPM for conditional image generation, specifically image inpainting, in diffusion models. Previous methods usually use a hard pixel reset or gradient backward to enforce the image to be coherent with the input image for image inpainting using pre-trained diffusion models. This paper, on the other hand, attempts to utilize the TPM along with the diffusion models to enforce the image to be coherent. At each timestep, the proposed method reconstructs the $x_0$ with both the diffusion model and TPM and takes the geometric mean to get the output. Results are compared on multiple datasets including CelebA, LSUN-Bedroom, and ImageNet.

**Strengths:**

1. The use of TPM for the conditional generation of diffusion models is interesting.
2. The overall quantitative results look good compared to previous methods.

**Weaknesses:**

1. The comparison only contains six different masks. In real application, the cases where the images are masked by some texts or patterns are also very common. It would be ideal to see more comparisons of such masks in arbitrary shapes.
2. The table only contains LPIPS for quantitative measurement, however, as image inpainting is an ill-posed problem, a user study would be beneficial in this case as previous works such as [1][2] perform.

[1] Towards Coherent Image Inpainting Using Denoising Diffusion Implicit Models\
[2] Repaint: Inpainting using denoising diffusion probabilistic models

**Questions:**

From Figure 1 and Table 3, there are some hyper-parameters specifically tuned for different datasets especially $t_{cut}$ which has also been applied in many previous works, could the authors provide an ablation study over the selection of the hyper-parameters?

---

> ### Author Response · Authors · 2023-11-19
> **Response to Reviewer yw5k (part 1)**
>
> We thank the reviewer for their insightful comments.
>
> > It would be ideal to see more comparisons of such masks in arbitrary shapes
>
> Thank you for the suggestion. We additionally adopted a set of randomly generated “wide” masks following [1] and [2]. The masks are generated by uniformly sampling from polygonal chains dilated by a high random width and rectangles of arbitrary aspect ratios. We downloaded a set of 100 such masks provided by [1] and ran all algorithms with the same set of hyperparameters used in other tasks (no hyperparameter tuning specific to these masks is done). We have added the full results in the updated paper (Table 1) and below is a summary of the new results:
>
> |    Dataset & Mask &nbsp;&nbsp;&nbsp;  | Tiramisu |  CoPaint  |  RePaint  |  DDNM  |  DDRM  |  &nbsp; DPS   &nbsp;  | Resampling |
>
> | CelebAHQ & Wide  |  &nbsp; 0.069 &nbsp; &nbsp;  |   0.072   &nbsp;&nbsp; |   &nbsp; 0.075  &nbsp;  |  &nbsp; 0.112  &nbsp; |   0.132 &nbsp;  |  0.078    |   &nbsp;  0.128     &nbsp; &nbsp;&nbsp;&nbsp;&nbsp;&nbsp; |
>
> | ImageNet & Wide  |  &nbsp;  0.125 &nbsp; &nbsp;  |    0.128  &nbsp;&nbsp; |   &nbsp;  0.127 &nbsp;  |  &nbsp;   0.198 &nbsp; |  0.197  &nbsp;  |  0.132 |   &nbsp;  0.196 &nbsp; &nbsp;&nbsp;&nbsp;&nbsp;&nbsp; |
>
> | LSUN & Wide &nbsp;  &nbsp;  &nbsp;  &nbsp;    |  &nbsp;    0.116  &nbsp; &nbsp;  |     0.115 &nbsp;&nbsp; |   &nbsp;   0.124 &nbsp;  |  &nbsp;   0.135 &nbsp; |  0.204  &nbsp;  |  0.108  |   &nbsp;   0.202 &nbsp; &nbsp;&nbsp;&nbsp;&nbsp;&nbsp; |
>
> Tiramisu achieves the best LPIPS score on CelebA-HQ and ImageNet, and is behind DPS in LSUN-Bedroom. Overall, Tiramisu achieves the best LPIPS score on 14 out of 21 tasks. Since the LPIPS scores alone may not fully justify the performance of Tiramisu, please also refer to the response to the next question (about user study).
>
> > a user study would be beneficial in this case as previous works such as [1][2] perform
>
> We conduct a user study based on all three datasets and two masks (“Expand1” and the newly added “Wide” masks adopted from [1,2]). Detailed settings and results are added in the updated paper (Appendix E.1). We copy the result table from the paper:
>
> | &nbsp;&nbsp;&nbsp;&nbsp; Dataset & Mask     &nbsp;&nbsp; &nbsp;  |  &nbsp;  Tiramisu &nbsp; | CoPaint | RePaint | DPS |
>
> | CelebAHQ & Expand1  |  Reference    |   &nbsp;&nbsp;&nbsp;&nbsp;  22    &nbsp;&nbsp;&nbsp;&nbsp;  |    &nbsp;&nbsp;&nbsp;&nbsp; 34   &nbsp;&nbsp;&nbsp;&nbsp;  |  &nbsp;14 &nbsp; |
>
> | CelebAHQ & Wide  &nbsp; &nbsp; &nbsp; |  Reference    |   &nbsp;&nbsp;&nbsp;&nbsp;  26    &nbsp;&nbsp;&nbsp;&nbsp;  |    &nbsp;&nbsp;&nbsp;&nbsp; 30   &nbsp;&nbsp;&nbsp;&nbsp;  |  &nbsp; 22 &nbsp; |
>
> | ImageNet & Expand1  |  Reference    |   &nbsp;&nbsp;&nbsp;&nbsp;  32    &nbsp;&nbsp;&nbsp;&nbsp;  |    &nbsp;&nbsp;&nbsp;&nbsp; 20   &nbsp;&nbsp;&nbsp;&nbsp;  |  &nbsp; 24 &nbsp; |
>
> | ImageNet & Wide &nbsp;&nbsp; &nbsp;&nbsp;    |  Reference    |   &nbsp;&nbsp;&nbsp;&nbsp;  14    &nbsp;&nbsp;&nbsp;&nbsp;  |    &nbsp;&nbsp;&nbsp;&nbsp;&nbsp; 6   &nbsp;&nbsp;&nbsp;&nbsp;&nbsp;  |  &nbsp; 12 &nbsp; |
>
> | LSUN & Expand1&nbsp; &nbsp;&nbsp; &nbsp; &nbsp;    |  Reference    |   &nbsp;&nbsp;&nbsp;&nbsp;  18    &nbsp;&nbsp;&nbsp;&nbsp;  |    &nbsp;&nbsp;&nbsp;&nbsp;&nbsp; 2   &nbsp;&nbsp;&nbsp;&nbsp;&nbsp;  |  &nbsp; 8 &nbsp; |
>
> | LSUN & Wide &nbsp;&nbsp; &nbsp; &nbsp; &nbsp; &nbsp; &nbsp; &nbsp;   |  Reference    |   &nbsp;&nbsp;&nbsp;&nbsp; &nbsp; 4    &nbsp;&nbsp;&nbsp;&nbsp; &nbsp; |    &nbsp;&nbsp;&nbsp;&nbsp;&nbsp; 6   &nbsp;&nbsp;&nbsp;&nbsp;&nbsp;  |  &nbsp; -6 &nbsp; |
>
>
> We use the vote difference (%) metric, which is the percentage of votes to Tiramisu subtracted by that of the baseline. A positive vote difference value means images generated by Tiramisu are preferred compared to the baselines, while a negative value suggests that the baseline is better than Tiramisu.
>
> We compare against the three strongest baselines (CoPaint, RePaint, and DPS) as their average LPIPS scores are not significantly worse than Tiramisu. As shown in the table, in all but one task the vote difference score is positive (and often quite large), which indicates the superiority of Tiramisu compared to the baselines.

---

> > ### Author Response · Authors · 2023-11-19
> > **Response to Reviewer yw5k (part 2)**
> >
> > > there are some hyper-parameters specifically tuned for different datasets especially t_cut which has also been applied in many previous works, could the authors provide an ablation study over the selection of the hyper-parameters?
> >
> > We provide an ablation study over the value of t_cut in the updated paper (Section 6.2, the “computational efficiency” paragraph). We adopt the “CelebA-HQ + the Expand1 mask” task. All other hyperparameters are kept the same and we change t_cut (we used “% denoising steps using PC”, which has a one-to-one correspondence with t_cut). As shown in Figure 4 of the updated paper (for convenience, we copied the results in the figure to the table below), as we engage the PC in more denoising steps, the LPIPS score first decreases and then increases. Therefore, incorporating PCs in a moderate amount of steps gives the best performance (around 20% in this case).
> >
> > One explanation for the diminishing performance gain when using PCs in too many denoising steps is that the later stages of denoising are mainly responsible for refining the details in the image. And the guidance of PCs could be less effective in this aspect.
> >
> > | % denoising steps using PC  |   &nbsp;&nbsp;&nbsp;  5    &nbsp;&nbsp;&nbsp;&nbsp;|&nbsp;&nbsp;&nbsp;   10    &nbsp;&nbsp;&nbsp;|&nbsp; &nbsp;&nbsp;   20   &nbsp;&nbsp;&nbsp;|&nbsp;&nbsp;&nbsp;    30   &nbsp;&nbsp;&nbsp;|&nbsp;&nbsp;&nbsp;   40    &nbsp;&nbsp;&nbsp;|&nbsp;&nbsp;&nbsp;  50 &nbsp;&nbsp; |
> >
> > | &nbsp;&nbsp;&nbsp;&nbsp;&nbsp;&nbsp;&nbsp;&nbsp;&nbsp;&nbsp;&nbsp;&nbsp;&nbsp;&nbsp;&nbsp;&nbsp;&nbsp;&nbsp;&nbsp; LPIPS &nbsp;&nbsp;&nbsp;&nbsp;&nbsp;&nbsp;&nbsp;&nbsp;&nbsp;&nbsp;&nbsp;&nbsp;&nbsp;&nbsp;&nbsp;&nbsp;&nbsp;&nbsp;&nbsp; |  0.470 | 0.459 | &nbsp; 0.454  | 0.456 | 0.464 &nbsp; | 0.476 |
> >
> > | &nbsp;&nbsp;&nbsp;&nbsp;&nbsp;&nbsp;&nbsp;&nbsp;&nbsp;&nbsp;&nbsp;&nbsp;&nbsp;&nbsp;&nbsp;&nbsp; Runtime &nbsp;&nbsp;&nbsp;&nbsp;&nbsp;&nbsp;&nbsp;&nbsp;&nbsp;&nbsp;&nbsp;&nbsp;&nbsp;&nbsp;&nbsp;&nbsp; |  &nbsp; 109 &nbsp;  | &nbsp; 111 &nbsp;  |  &nbsp; 115 &nbsp;  |  &nbsp; 118 &nbsp; |  &nbsp;121 &nbsp;  | &nbsp; 125 &nbsp; |
> >
> > [1] Andreas Lugmayr, Martin Danelljan, Andres Romero, Fisher Yu, Radu Timofte, and Luc Van Gool. Repaint: Inpainting using denoising diffusion probabilistic models. In Proceedings of the IEEE/CVF Conference on Computer Vision and Pattern Recognition, pp. 11461–11471, 2022.
> >
> > [2] Roman Suvorov, Elizaveta Logacheva, Anton Mashikhin, Anastasia Remizova, Arsenii Ashukha, Aleksei Silvestrov, Naejin Kong, Harshith Goka, Kiwoong Park, and Victor Lempitsky. Resolutionrobust large mask inpainting with fourier convolutions. In Proceedings of the IEEE/CVF winter conference on applications of computer vision, pp. 2149–2159, 2022.

---

> > > ### Author Response · Authors · 2023-11-23
> > >
> > > Dear Reviewer yw5k,
> > >
> > > We were wondering if you have had a chance to read our reply to your feedback. As the time window for the rebuttal is closing soon, please let us know if there are any additional questions we can answer.
> > >
> > > Best,
> > >
> > > Authors

---

### Meta-Review · Area_Chair_eL33 · 2023-12-06

**Metareview:**

Three out of four reviewers found the paper marginally above the acceptance threshold. Reviewer dUar gave a higher rating of "6: marginally above the acceptance threshold" during the discussion with authors although the updated score is not reflected in his/her main review. Reviewers generally acknowledge the novelty of the approach. However, its limited scope in terms of mask variety and the lack of a comprehensive user study or additional metrics like FID, U-IDS are noted weaknesses. Authors addressed these concerns during the rebuttal and discussion period.

**Justification For Why Not Higher Score:**

The expressiveness of PCs is limited to images of small resolution. The proposed approach can lead to worse performance given too many denoising steps and may be less effective when dealing with small-hole inpainting.

**Justification For Why Not Lower Score:**

The integration of TPMs with diffusion models for image inpainting is a fresh perspective in the field. The authors have satisfactorily addressed the reviewers' concerns during discussions.

---

### Decision · Program_Chairs · 2024-01-16

Accept (poster)